# Lamella-heterostructured nanoporous bimetallic iron-cobalt alloy/oxyhydroxide and cerium oxynitride electrodes as stable catalysts for oxygen evolution

Shu-Pei Zeng[1,2], Hang Shi [1,2], Tian-Yi Dai[1,2], Yang Liu[1], Zi Wen [1], Gao-Feng Han [1], Tong-Hui Wang[1], Wei Zhang [1], Xing-You Lang [1]✉, Wei-Tao Zheng [1] & Qing Jiang [1]✉

Developing robust nonprecious-metal electrocatalysts with high activity towards sluggish oxygen-evolution reaction is paramount for large-scale hydrogen production via electrochemical water splitting. Here we report that self-supported laminate composite electrodes composed of alternating nanoporous bimetallic iron-cobalt alloy/oxyhydroxide and cerium oxynitride ($FeCo/CeO_{2-x}N_x$) heterolamellas hold great promise as highly efficient electrocatalysts for alkaline oxygen-evolution reaction. By virtue of three-dimensional nanoporous architecture to offer abundant and accessible electroactive $CoFeOOH/CeO_{2-x}N_x$ heterostructure interfaces through facilitating electron transfer and mass transport, nanoporous $FeCo/CeO_{2-x}N_x$ composite electrodes exhibit superior oxygen-evolution electrocatalysis in 1 M KOH, with ultralow Tafel slope of ~33 mV dec$^{-1}$. At overpotential of as low as 360 mV, they reach >3900 mA cm$^{-2}$ and retain exceptional stability at ~1900 mA cm$^{-2}$ for >1000 h, outperforming commercial $RuO_2$ and some representative oxygen-evolution-reaction catalysts recently reported. These electrochemical properties make them attractive candidates as oxygen-evolution-reaction electrocatalysts in electrolysis of water for large-scale hydrogen generation.

Electrochemical water splitting powered by renewable electricity from plentiful solar and wind resources is an attractive energy conversion technology for clean and large-scale hydrogen production[1–4]. It promises an environmentally friendly energy framework of water cycle by making use of molecular hydrogen ($H_2$) as a clean and high-density energy carrier to meet future global energy needs[4–7]. However, water electrolysis no matter in alkaline water electrolyzers or proton-exchange-membrane water electrolyzers persistently undergoes a low energy efficiency primarily due to sluggish kinetics of oxygen evolution reaction (OER) and insufficient activity of state-of-the-art

OER electrocatalysts[8–10]. This dilemma persecutes widespread implementation of electrochemical water-splitting technologies, especially for mass production of $H_2$ in industry, where they are required to deliver high current densities of >500 mA cm$^{-2}$ at a low overpotential of <300 mV over thousands of hours[11,12]. In this regard, it is of high desire to develop highly active, robust, and cost-effective OER electrocatalytic materials for highly efficient practical industrial electrolyzers. In view of the multi-proton-electron coupled OER taking place on the solid-liquid-gaseous interface of catalytic active sites[13–16], ideal anodic materials should comprise adequate electroactive sites with

[1]Key Laboratory of Automobile Materials (Jilin University), Ministry of Education, School of Materials Science and Engineering, and Electron Microscopy Center, Jilin University, Changchun 130022, China. [2]These authors contributed equally: Shu-Pei Zeng, Hang Shi, Tian-Yi Dai. ✉e-mail: xylang@jlu.edu.cn; jiangq@jlu.edu.cn

high intrinsic activity to boost water oxidation reaction[3,17], in addition to a rational and steady electrode structure that simultaneously facilitates electron transfer and mass transportation of $OH^-$ ions and $H_2O$/$O_2$ molecules to/from sufficient available electroactive sites, and withstands violent gas evolution[18–20]. Despite judicious engineering of benchmarking precious-metal OER electrocatalysts such as $RuO_2$ and $IrO_2$ on steady conductive supports could satisfy these versatile requirements[16,21,22], their scarcity, high cost and inferior durability substantially hamper the practical applications[23,24]. To replace precious metal electrocatalysts, many bimetallic or multi-metallic compounds based on Earth-abundant 3$d$ transition metals[25–27], such as iron (Fe), cobalt (Co), and nickel (Ni) have been explored extensively[28–32]. These include their oxides/hydroxides[3,8,9,25–34], oxide perovskites[25,26,35], sulfides/selenides[3,8,25,26,36–39], phosphides/nitrides[40,41] and molecular complexes[42] with impressive OER electrocatalytic behaviors in alkaline media. However, most of them work well only at low current densities (<100 mA cm$^{-2}$) for dozens of hours, unsatisfying the industrial requirements of practical electrolyzers[10,24–32]. This is probably because of their insufficient intrinsic activity of electroactive sites with too strong or too weak adsorption energies of *OH, *O, and *OOH intermediates[3,9,10,17], as well as poor accessibility of electroactive sites and/or low electron transferability in industrial-scale electrode materials[18–20], particularly for the ones made of traditionally low-dimensional nanocatalysts, which have to be casted on current collectors using polymer binder to maintain their electrical contact[18–20,27,43,44]. Therein, these nanocatalysts stack densely to inevitably bury most electroactive sites, inhibit mass transportation, and bring supplementary interfaces with high contact resistance, all of which lead to substantial overpotentials at the required high current densities[18–20,31,45]. Furthermore, these casted nanocatalysts usually suffer from inferior durability because they are apt to be peeled off from the substrates due to their weak adhesive force[18–20,45].

Here we report self-supported laminate composite electrodes of alternating hierarchical nanoporous bimetallic iron-cobalt alloy/oxyhydroxide and cerium oxynitride (FeCo/$CeO_{2-x}N_x$) heterolamellas as highly efficient and robust alkaline OER electrocatalytic materials. Owing to the constituent $CeO_{2-x}N_x$ that not only properly modulates the contiguous CoFeOOH to have near-optimal adsorption energies of *OH, *O and *OOH intermediates but also immediately adsorbs the produced oxygen by making use of its exceptional oxygen-storage capability, the CoFeOOH/$CeO_{2-x}N_x$ interfaces serve as the electroactive sites with remarkably enhanced OER activity (~0.558 mA cm$^{-2}_{ECSA}$ at overpotential of 360 mV). By virtue of three-dimensional bicontinuous nanoporous architecture offering abundant electroactive sites of CoFeOOH/$CeO_{2-x}N_x$ heterostructure through facilitating electron transfer and mass transport, the nanoporous FeCo/$CeO_{2-x}N_x$ laminate composite electrodes exhibit superior alkaline OER electrocatalysis, with low onset overpotential (~186 mV) and Tafel slope (~33 mV dec$^{-1}$), in 1 M KOH solution. They reach ultrahigh current densities of >3900 mA cm$^{-2}$ only at a low overpotential of 360 mV. Furthermore, they maintain exceptional stability even at the current density of as high as ~1900 mA cm$^{-2}$ for more than 1000 h. These impressive electrochemical properties outperform those of commercially available $RuO_2$ and some representative OER catalysts recently reported and enlist the nanoporous FeCo/$CeO_{2-x}N_x$ electrodes to hold great promise as attractive OER catalysts for large-scale hydrogen generation via water electrolysis.

## Results

### Preparation and physicochemical characterizations of nanoporous laminate composite electrodes

The self-supported nanoporous FeCo/$CeO_{2-x}N_x$ laminate composite electrodes are fabricated by alloying/dealloying of lamella-nanostructured eutectic intermetallic compounds, followed by a thermal nitridation procedure (Fig. 1a). Briefly, molten alloys of $Fe_{25-y-z}Co_yCe_zAl_{75}$ (at%, where $y$, $z = 0$, 5 or 25) are firstly made by arc-melting pure Fe and Al metals with/without the addition of Co and/or Ce (Supplementary Fig. 1), of which the composition ratio ensures the formation of single- or multi-phase nanostructures of precursor alloys. When cooled to ambient temperature in a water cycle-assisted furnace, there takes place eutectic solidification reaction in the representative precursor alloy of $Fe_{25-y-z}Co_yCe_zAl_{75}$ with $y = z = 5$, i.e., $Fe_{15}Co_5Ce_5Al_{75}$, to symbiotically form lamellar dual-phase nanostructured intermetallic $Al_{13}(FeCo)_4$ and $Al_{11}Ce_3$ eutectoids[46–49]. As demonstrated by its X-ray diffraction (XRD) characterization (Supplementary Fig. 2), there mainly appear two sets of characteristic patterns attributed to the monoclinic $Al_{11}Ce_3$ (JCPDS 19-0006) and the orthorhombic $Al_{13}Fe_4$ (JCPDS 29-0042) intermetallic compounds, respectively. Relative to the standard line patterns of intermetallic $Al_{13}Fe_4$, the diffraction peaks of $Al_{13}(FeCo)_4$ shift to higher angles, indicating the incorporation of Co component in the intermetallic $Al_{13}Fe_4$ matrix[48]. After cut into sheets with thickness of ~400 μm, these eutectic $Fe_{15}Co_5Ce_5Al_{75}$ sheets are immersed in $N_2$-purged aqueous KOH solution to selectively etch less-noble Al component via a chemical dealloying process[50,51], during which the dissolution of Al from the symbiotic intermetallic $Al_{13}(FeCo)_4$ and $Al_{11}Ce_3$ phases enables monolithic nanoporous bimetallic FeCo alloy/oxide and cerium hydroxide/oxide (Ce-O) (FeCo/Ce-O) laminate composites by making use of the distinct chemical activities of Fe, Co and Ce elements (Supplementary Figure 3)[52]. The as-dealloyed nanoporous FeCo/Ce-O sheets are further annealed at 600 °C in Ar/$NH_3$ atmosphere for the preparation of nanoporous FeCo/$CeO_{2-x}N_x$ composite electrodes, wherein the constituent Ce-O lamellas evolves into the N-doped $CeO_2$ ($CeO_{2-x}N_x$) ones during the thermal nitridation. For comparison, the corresponding individuals, i.e., nanoporous bimetallic FeCo alloy/oxide and nanoporous $CeO_{2-x}N_x$, are also prepared by the same procedure on the basis of their single-phase precursor alloys of $Fe_{25-y-z}Co_yCe_zAl_{75}$ with $y = 5$, $z = 0$ (i.e., $Fe_{20}Co_5Al_{75}$) and $y = 0$, $z = 25$ (i.e., $Ce_{25}Al_{75}$), respectively (Supplementary Figure 1, 2).

Figure 1b and Supplementary Figure 4 show representative scanning electron microscope (SEM) images of as-prepared nanoporous FeCo/$CeO_{2-x}N_x$ composite electrode, displaying the unique architecture consisting of periodically alternating nanoporous FeCo alloy/oxide and $CeO_{2-x}N_x$ lamellas with the interlamellar spacing of ~600 nm. XRD characterization of nanoporous FeCo/$CeO_{2-x}N_x$ verifies the composite structure with two sets of diffraction patterns at 2θ = 44.6°, 64.7°, 82.4° and 28.5°, 33.0°, 47.4°, 56.3°, which correspond to the (110), (200), (211) planes of body-centered cubic (bcc) Fe (JCPDS 01-1262) and the (111), (200), (220) and (311) planes of face-centered cubic (fcc) $CeO_2$ (JCPDS 04-0593) (Fig. 1c), respectively[19,53]. Relative to the standard line patterns of Fe, the diffraction peaks of the constituent FeCo alloy shift to higher angles; while for the $CeO_{2-x}N_x$, the characteristic peaks shift to lower angles compared to those of $CeO_2$. The amount of surface oxide is too little to be detected by XRD. Figure 1d shows a typical high-resolution transmission electron microscope (HRTEM) image of FeCo/$CeO_{2-x}N_x$ interface, in which the constituent FeCo alloy and $CeO_{2-x}N_x$ compound viewed along their <110> and <112> zone axis, respectively, are identified by their corresponding fast Fourier transform (FFT) patterns (Fig. 1e, f) and selected area electron diffraction (SAED) patterns (inset of Fig. 1d). Therein, the constituent FeCo alloy is in bcc Fe phase (Fig. 1e) and the $CeO_{2-x}N_x$ compound is in fcc $CeO_2$ phase (Fig. 1f). Despite their distinct crystallographic structures, they are highly coherent with each other. In the FeCo and $CeO_{2-x}N_x$ regions in HRTEM image (Fig. 1d), their interplanar spacings of ~0.198 nm and ~0.313 nm correspond to the lattice planes of Fe(110) and $CeO_2$(111) but deviate from the corresponding standard values (0.203 nm and 0.312 nm) due to the incorporation of Co and N, respectively. In the SAED pattern, there display clear diffraction rings matching mostly with the fcc $CeO_{2-x}N_x$ (111), (200), (220), (311) and bcc

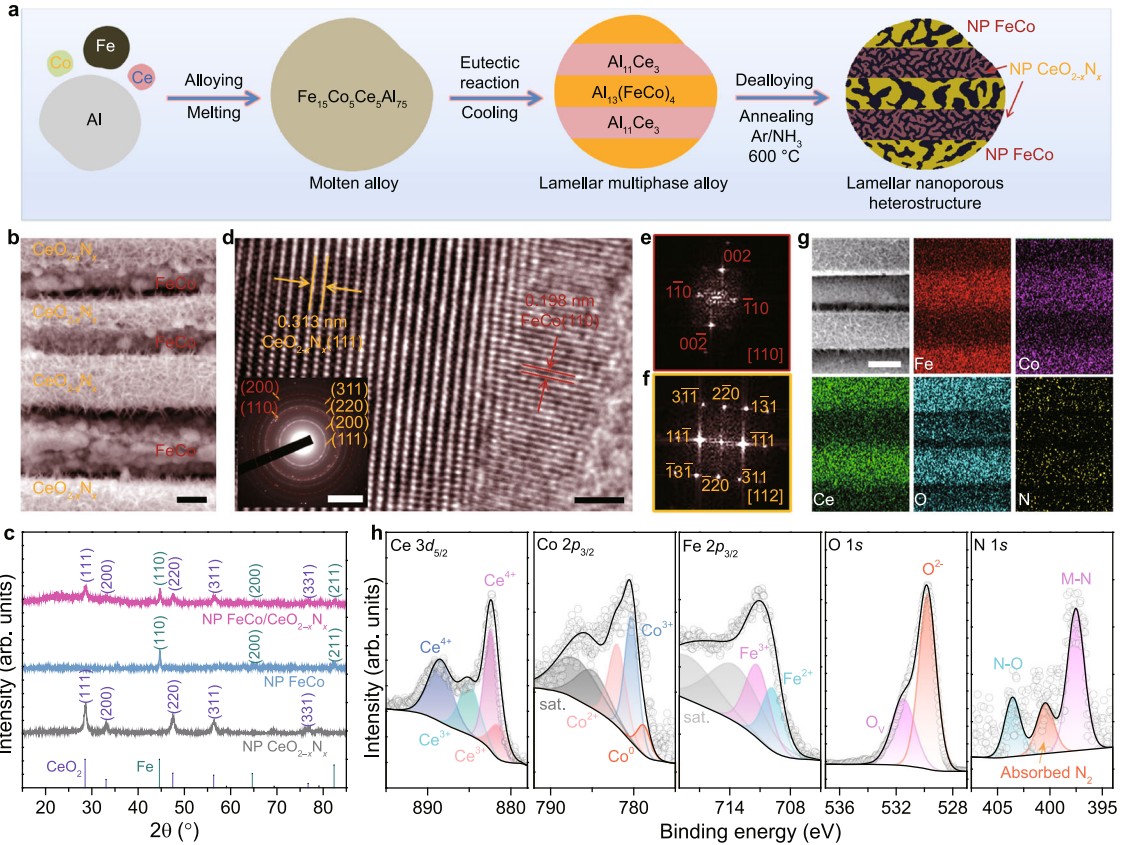

**Fig. 1 | Preparation scheme and microstructure of nanoporous FeCo/CeO$_{2-x}$N$_x$ electrodes. a** Schematic diagram to illustrate fabrication procedure of lamellar nanoporous (NP) FeCo/CeO$_{2-x}$N$_x$ composite electrode, during which precursor Fe$_{15}$Co$_5$Ce$_5$Al$_{75}$ eutectic alloy composed of lamella-nanostructured Al$_{11}$Ce$_3$ and Al$_{13}$(FeCo)$_4$ intermetallic compounds are prepared by solidification eutectic reaction and then chemically dealloyed in N$_2$-purged KOH solution to remove less-noble Al, followed by a thermal nitrification process at 600 °C in Ar/NH$_3$ atmosphere. **b** Top-view SEM image of NP FeCo/CeO$_{2-x}$N$_x$ composite electrode, displaying an architecture composed of periodically alternating hierarchical nanoporous FeCo alloy and nanoporous CeO$_{2-x}$N$_x$ lamellas. Scale bar, 200 nm. **c** XRD patterns of NP FeCo/CeO$_{2-x}$N$_x$ composite electrode (pink), as well as its

corresponding NP FeCo alloy (blue) and NP CeO$_{2-x}$N$_x$ individuals (grey). The line patterns show reference cards 04-0593 and 01-1262 for face-centered cubic CeO$_2$ and body-centered cubic Fe metal according to JCPDS. **d** Representative HRTEM image of heterostructured FeCo/CeO$_{2-x}$N$_x$ interface. Scale bar, 1 nm. Inset: SAED patterns of FeCo/CeO$_{2-x}$N$_x$ heterostructure. Scale bar, 5 1/nm. **e, f** FFT patterns of FeCo alloy (**e**) and CeO$_{2-x}$N$_x$ (**f**) in the selected areas in FeCo/CeO$_{2-x}$N$_x$ heterostructure in (**d**). **g** Typical SEM image and the corresponding EDS elemental mapping images of NP FeCo/CeO$_{2-x}$N$_x$ electrode for Fe, Co, Ce, O, and N elements. Scale bar, 200 nm. **h** High-resolution XPS spectra of Ce 3$d$, Co 2$p$, Fe 2$p$, O 1$s$, and N 1$s$ on the surface of NP FeCo/CeO$_{2-x}$N$_x$ electrode.

FeCo (110), (200) planes (inset of Fig. 1d), respectively, which illustrates the heterojunction of FeCo/CeO$_{2-x}$N$_x$. The laminate composite structure of nanoporous FeCo/CeO$_{2-x}$N$_x$ is also illustrated by its SEM energy-dispersive X-ray spectroscopy (EDS) elemental mappings (Fig. 1g), where Fe, Co and Ce, O, N atoms periodically distribute in nanoporous FeCo/CeO$_{2-x}$N$_x$, depending on the presence alternating nanoporous FeCo alloy/oxide and CeO$_{2-x}$N$_x$ compound lamellas. Owing to the residue of Ce component in the FeCo alloy lamellas caused by eutectic solidification reaction of precursor alloy, there also observe some Ce atoms along with O and N to distribute in the constituent nanoporous FeCo alloy/oxide lamellas. This enables abundant FeCo/CeO$_{2-x}$N$_x$ heterointerfaces during the chemical dealloying and thermal nitridation processes (Supplementary Fig. 5), in addition to the ones located between nanoporous FeCo alloy/oxide and CeO$_{2-x}$N$_x$ lamellas. The distribution of less O atoms along the nanoporous FeCo alloy lamellas is due to the hereditary surface oxide from the Co incorporated Fe$_2$O$_3$ layer, i.e., Co-Fe$_2$O$_3$, which in-situ grows on the surface of FeCo alloy ligaments during the chemical dealloying (Supplementary Figure 3b) and transforms into the Co-Fe$_3$O$_4$ in the subsequent thermal nitridation process. Owing to the transformation of Co-Fe$_2$O$_3$ into Co-Fe$_3$O$_4$, there appear two neoformative Raman bands in the Raman spectrum of nanoporous FeCo/CeO$_{2-x}$N$_x$, with the sharp peaks at ~294 and ~672 cm$^{-1}$ assigned to the $E_g$ and $A_{1g}$ vibrational

modes of Co-Fe$_3$O$_4$, while the characteristic ones at ~212 and at ~274 cm$^{-1}$ for the $A_{1g}$ and 2$E_g$ vibrational modes of Co-Fe$_2$O$_3$ disappear (Supplementary Figure 6)[54,55]. In addition, the Raman band at ~458 cm$^{-1}$ is attributed to the $F_{2g}$ vibrational mode of CeO$_{2-x}$N$_x$[56], with an evident redshift relative to that of the pristine fluorite-structured CeO$_2$ (~442 cm$^{-1}$) because of the N doping (Supplementary Figure 6)[57]. X-ray photoelectron spectroscopy (XPS) analysis further verifies the presence of Ce, O and N atoms, in addition to Fe and Co atoms with an atomic ratio of 75: 25, in the nanoporous FeCo/CeO$_{2-x}$N$_x$ composite electrode (Supplementary Figure 7). The Co 2$p$ and Fe 2$p$ XPS spectra show that the surface Co and Fe atoms of nanoporous FeCo/CeO$_{2-x}$N$_x$ electrode are mainly in the oxidized states of Co$^{3+}$, Co$^{2+}$ and Fe$^{3+}$, Fe$^{2+}$ due to the presence of surface Co-Fe$_3$O$_4$ layer on the FeCo lamellas (Fig. 1h and Supplementary Fig. 6). Owing to the transformation of Ce(OH)$_3$ to CeO$_2$ along with the N doping, the surface Ce atoms of nanoporous FeCo/CeO$_{2-x}$N$_x$ are in the mixed chemical states of Ce$^{4+}$ and Ce$^{3+}$ with the Ce$^{4+}$/Ce$^{3+}$ ratio increasing to 76/24 from 23/77 in nanoporous FeCo/Ce-O (Fig. 1h and Supplementary Fig. 8a)[58,59]. The N 1$s$ XPS spectrum with the M-N characteristic peak at 397.5 eV demonstrates the formation of Ce-N bonds, in addition to the ones at 400.5 and 403.6 eV corresponding to the absorbed nitrogen and the O-N, respectively[60]. Because of the N doping, this transformation also gives rise to a high concentration of oxygen defects (O$_V$). As shown in

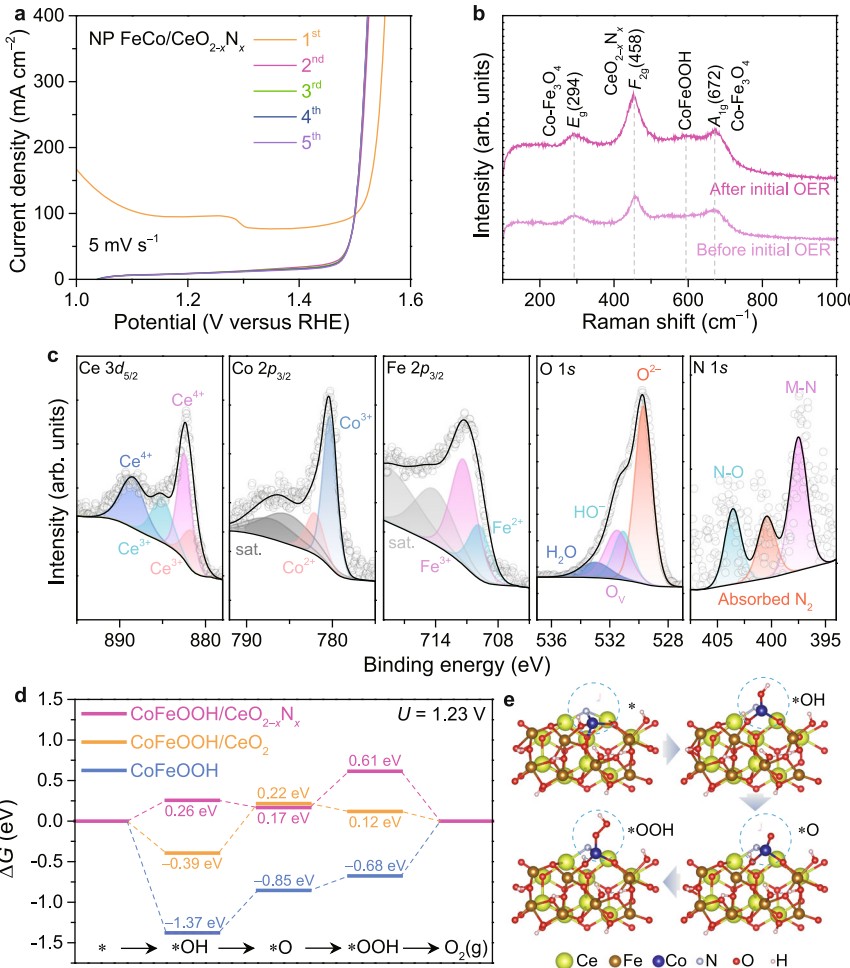

**Fig. 2 | Surface reconstruction of composite electrode. a** Typical initial five OER polarization curves of nanoporous (NP) FeCo/CeO$_{2-x}$N$_x$ composite electrode. Scan rate: 5 mV s$^{-1}$. **b** Raman spectra of NP FeCo/CeO$_{2-x}$N$_x$ composite electrode before (light pink) and after (dark pink) initial six OER polarization curves in 1 M KOH aqueous electrolyte. **c** High-resolution XPS spectra of Ce 3$d_{5/2}$, Co 2$p_{3/2}$, Fe 2$p_{3/2}$, O 1$s$, and N 1$s$ in NP FeCo/CeO$_{2-x}$N$_x$ composite electrode after measurement of six OER polarization curves. **d** Free energy diagrams for OER at 1.23 V bias over CoFeOOH/CeO$_{2-x}$N$_x$ (pink), CoFeOOH/CeO$_2$ interface (yellow), and CoFeOOH surface (blue). **e** Interface configuration of CoFeOOH/CeO$_{2-x}$N$_x$ heterostructure at four different stages during the OER electrocatalysis.

the O 1$s$ XPS spectrum, there are two different oxygen species, i.e., the O$^{2-}$ in the M-O bonds and the oxygen defects O$_V$, to correspond to the peaks at the binding energies of 529.7 and 531.7 eV (Fig. 1h)[53].

## Surface reconstruction of nanoporous laminate composite electrodes

When serving as the alkaline OER catalyst, the nanoporous FeCo/CeO$_{2-x}$N$_x$ is prone to undergo surface reconstruction into (oxy) hydroxides under the strongly oxidative environment[18,29,61–64]. This is attested by the quite different voltammetric behaviors of nanoporous FeCo/CeO$_{2-x}$N$_x$ in the initial linear scanning voltammetry (LSV) polarizations in 1 M KOH electrolyte. As shown in Fig. 2a, the nanoporous FeCo/CeO$_{2-x}$N$_x$ exhibits an evident cathodic peak and high pseudocapacitance in the first LSV curve compared with the subsequent ones, where the pseudocapacitive behavior completely disappears. This is probably due to the irreversible surface oxidation of Co-Fe$_3$O$_4$ to form stable oxyhydroxide (CoFeOOH) on the constituent FeCo[41,53,65]. While for the CeO$_{2-x}$N$_x$, the N doping enlists the constituent CeO$_{2-x}$N$_x$ to maintain exceptional electrochemical stability via the formation of Ce-N bonds (Supplementary Figure 9). These conjectures are verified by Raman and XPS characterizations of nanoporous FeCo/CeO$_{2-x}$N$_x$ before and after the initial OER test, in which the polarization curves are performed for six cycles. As shown in Fig. 2b, the nanoporous FeCo/CeO$_{2-x}$N$_x$ after the initial OER test displays the almost same

Raman spectrum as the pristine one except for the neoformative characteristic Raman band at ~590 cm$^{-1}$, which is assigned to the $E_g$ of CoFeOOH[53]. The weak Raman signal at ~590 cm$^{-1}$ indicates that there only forms an ultrathin CoFeOOH layer on the surface of nanoporous FeCo alloy/oxide ligaments, which is conducive to their electron transportation from electroactive sites to current collectors during the OER processes. Figure 2c presents high-resolution XPS spectra of Ce 3$d$, Co 2$p$, Fe 2$p$, O 1$s$, and N 1$s$ for the nanoporous FeCo/CeO$_{2-x}$N$_x$ after the initial OER test. Owing to the surface oxidation, the surface Co and Fe atoms are fully in the oxidized states of Co$^{3+}$, Co$^{2+}$ and Fe$^{3+}$, Fe$^{2+}$ with the Co$^{3+}$/Co$^{2+}$ and Fe$^{3+}$/Fe$^{2+}$ ratios increasing to 74/26 and 70/30, respectively, in contrast to the pristine FeCo/CeO$_{2-x}$N$_x$ with Co$^0$/Co$^{2+}$/ Co$^{3+}$ = 8/41/51 and Fe$^{3+}$/Fe$^{2+}$ = 58/42 (Supplementary Fig. 10). Nevertheless, there do not observe evident changes in the chemical states of surface Ce, O and N atoms on the nanoporous FeCo/CeO$_{2-x}$N$_x$ before and after the initial OER test (Fig. 2c and Supplementary Fig. 10). Especially for the capricious Ce atoms, they keep the constant Ce$^{4+}$/ Ce$^{3+}$ ratio of 75/25, different from those on as-dealloyed nanoporous FeCo/Ce-O (Supplementary Fig. 8a, b), where the chemical states of Ce atoms completely change to Ce$^{4+}$ from the Ce$^{4+}$ and Ce$^{3+}$ with a ratio of 23/77 after the initial OER test (Supplementary Figure 11). This fact suggests the significant role of N atoms in stabilizing the chemical state of Ce atoms for further modulating the contiguous CoFeOOH to have near-optimal adsorption energies of *OH, *O and *OOH intermediates.

As a result, the OER current densities in the subsequent LSV curves are much higher than that in the first LSV (Fig. 2a). The synergistic effects between CoFeOOH and $CeO_{2-x}N_x$ in nanoporous FeCo/$CeO_{2-x}N_x$ composite electrode is also illustrated by density functional theory (DFT) calculations based on CoFeOOH/$CeO_{2-x}N_x$ heterostructure that is constructed by combining amorphous O terminal CoFeOOH layer with the $CeO_2$ slab doped with/without N (Supplementary Fig. 12a, b). Bader charge analysis elucidates the change in the electronic structure after the incorporation of N. As shown in Supplementary Fig. 13a, b, there takes place more electron transfer from $CeO_{2-x}N_x$ to CoFeOOH in the CoFeOOH/$CeO_{2-x}N_x$ than from $CeO_2$ to CoFeOOH in CoFeOOH/$CeO_2$. Therein, the Co in CoFeOOH/$CeO_{2-x}N_x$ possesses a low atomic charge of $+1.18|e|$ compared with the Co atoms in CoFeOOH/$CeO_2$ ($+1.24|e|$) and bare CoFeOOH ($+1.45|e|$). This will weaken the adsorption energies of O species on CoFeOOH/$CeO_{2-x}N_x$ as a consequence of downshift of $d$-band center relative to CoFeOOH/$CeO_2$ and CoFeOOH (Supplementary Fig. 14). With an assumption that the adsorption of intermediates on electroactive Co atoms starts from the adsorption of $OH^-$ ion, followed by the sequential deprotonation to form *O, O-O bonding formation to generate *OOH, and desorption to produce oxygen, Fig. 2d shows the Gibbs free energy profiles ($\Delta G$) for the intermediates and products on CoFeOOH/$CeO_{2-x}N_x$, CoFeOOH/$CeO_2$ and bare CoFeOOH during the alkaline OER processes at $U = 1.23$ V versus RHE (Supplementary Fig. 12). Obviously, they suffer from different rate-determining step (RDS) due to their distinct adsorption capability of O species. When mediated by CoFeOOH/$CeO_{2-x}N_x$, the RDS of OER is the O-O bond formation via *O intermediate reacting with another $OH^-$ to form the *OOH, where the $\Delta G_{RDS}$ is as low as ~0.44 eV. This is different from the RDSs on CoFeOOH/$CeO_2$ (the deprotonation of *OH for the formation of *O, $\Delta G_{RDS} = \sim 0.61$ eV) and bare CoFeOOH (the $O_2$ desorption with $\Delta G_{RDS} = \sim 0.68$ eV). The lowest energy barrier enlists the CoFeOOH/$CeO_{2-x}N_x$ interface as the electroactive sites to substantially boost OER kinetics[30,39,53].

## Electrochemical characterizations of nanoporous laminate composite electrodes

To evaluate the electrocatalysis, all self-supported nanoporous electrode materials are directly used as working electrodes for electrochemical measurements, which are performed in $O_2$-saturated 1 M KOH electrolyte in a classic three-electrode cell with a graphite counter electrode and an Ag/AgCl reference electrode. According to the calibration experiment (Supplementary Fig. 15), all potentials are iR corrected and calibrated with respect to the reversible hydrogen electrode (RHE). Figure 3a shows typical OER polarization curve of nanoporous FeCo/$CeO_{2-x}N_x$ composite electrode, comparing with those of nanoporous FeCo/Ce-O, FeCo and $CeO_{2-x}N_x$ electrodes, as well as that of nickel foam supported $RuO_2$ ($RuO_2$/NF) and that of commercially available $RuO_2$ nanoparticles immobilized on glassy carbon electrode ($RuO_2$/GC) by dint of Nafion as polymer binder. Owing to the presence of abundant and electroactive CoFeOOH/$CeO_{2-x}N_x$ heterostructure, the nanoporous FeCo/$CeO_{2-x}N_x$ electrode exhibits significantly enhanced OER electrocatalytic behaviors compared with the nanoporous FeCo and $CeO_{2-x}N_x$ individuals. As shown in Fig. 3a, the nanoporous FeCo/$CeO_{2-x}N_x$ has an ultralow onset overpotential of ~186 mV, in sharp contrast with the nanoporous FeCo (~257 mV) and $CeO_{2-x}N_x$ (~374 mV) individual electrodes. As the overpotential increases to 360 mV, the OER current density of nanoporous FeCo/$CeO_{2-x}N_x$ electrode dramatically increases to ~3940 mA cm$^{-2}$, much higher than nanoporous FeCo (~152 mA cm$^{-2}$) and $CeO_{2-x}N_x$ electrodes (~2.55 mA cm$^{-2}$), respectively. Furthermore, this value is ~22- and ~35-fold higher than those of $RuO_2$/NF (~180 mA cm$^{-2}$) and $RuO_2$/GC (~114 mA cm$^{-2}$), respectively, although $RuO_2$ is usually expected as one of benchmark catalysts for the OER (Fig. 3b)[3,9,10,21,23]. Even in a mature technology of dimensionally stable anode (DSA), the Ti mesh-supported $RuO_2$ DSA exhibits much inferior electrocatalytic

behavior compared with nanoporous FeCo/$CeO_{2-x}N_x$ (Supplementary Figure 16). To demonstrate reproducibility, five nanoporous FeCo/$CeO_{2-x}N_x$ electrodes are prepared by the same procedure. Supplementary Figure 17 shows their OER polarization curves with high overlap, indicating excellent reproducibility. Whereas the nanoporous FeCo/Ce-O electrode has the almost same nanoporous architecture to simultaneously facilitate electron transfer and mass transportation (Supplementary Figure 3), it only achieves the current density of ~450 mA cm$^{-2}$, a ninth of the value of nanoporous FeCo/$CeO_{2-x}N_x$ electrode, at the overpotential of 360 mV. According to the double-layer capacitance measurements[66], the electrochemical surface area (ECSA) of nanoporous FeCo/$CeO_{2-x}N_x$ electrode is estimated to be ~0.79-fold of nanoporous FeCo/Ce-O electrode (Supplementary Figure 18), unable to account for the remarkable enhancement in OER current density. This fact indicates the important role of $CeO_{2-x}N_x$, instead of Ce-O, in improving the electrocatalytic activity of CoFeOOH via the formation of CoFeOOH/$CeO_{2-x}N_x$ heterostructure. Therein, the incorporation of N atoms weakens the adsorption energy of *OH intermediate on the electroactive Co atoms and results in the shift of RDS from the *O formation step on the CoFeOOH/$CeO_2$ to the *OOH generation step on the CoFeOOH/$CeO_{2-x}N_x$ (Fig. 2d, e). As a result, the specific activity of nanoporous FeCo/$CeO_{2-x}N_x$ is evaluated to reach as high as ~0.558 mA cm$^{-2}_{ECSA}$ at overpotential of 360 mV, more than 15-fold higher than that of nanoporous FeCo electrode with the electroactive CoFeOOH (~0.034 mA cm$^{-2}_{ECSA}$) (Supplementary Figure 19). The superior electrocatalytic activity of nanoporous FeCo/$CeO_{2-x}N_x$ electrode is also manifested by the ultralow Tafel slope of ~33 mV dec$^{-1}$, the smallest value among the investigated nanoporous electrodes, such as nanoporous FeCo (~68 mV dec$^{-1}$), $CeO_{2-x}N_x$ (~151 mV dec$^{-1}$) and FeCo/Ce-O (~71 mV dec$^{-1}$) as well as $RuO_2$/NF (~78 mV dec$^{-1}$) and $RuO_2$/GC (~93 mV dec$^{-1}$) (Fig. 3c). The substantially boosted kinetics of OER on the nanoporous FeCo/$CeO_{2-x}N_x$ is further revealed by electrochemical impedance spectroscopy (EIS) measurements. As shown in the Nyquist plots for nanoporous FeCo/$CeO_{2-x}N_x$, FeCo/Ce-O, FeCo, and $CeO_{2-x}N_x$ electrodes as well as $RuO_2$/NF and $RuO_2$/GC (Fig. 3d), their EIS spectra display two characteristic semicircles with distinct diameters in the middle- to low-frequency range, which correspond to different charge transfer resistances ($R_{CT}$) and the pore resistance ($R_P$) in parallel with the constant phase elements (CPEs). At the high frequencies, the intercept on the real axis represents the intrinsic resistance ($R_I$) of both the electrolyte and electrode. Based on these general descriptors in the equivalent circuit (inset of Fig. 3d), the $R_{CT}$ and $R_P$ values of nanoporous FeCo/$CeO_{2-x}N_x$ are as low as ~1.3 $\Omega$ and ~1.6 $\Omega$, respectively, demonstrating the superior reaction kinetics and mass transportation kinetics, in sharp contrast with nanoporous FeCo/Ce-O (~3.3 $\Omega$, ~1.6 $\Omega$), FeCo (~13.3 $\Omega$, ~1.9 $\Omega$) and $CeO_{2-x}N_x$ electrodes (~797 $\Omega$, ~12.6 $\Omega$), as well as the values of $RuO_2$/NF (~10.1 $\Omega$, ~2.8 $\Omega$) (Supplementary Fig. 20a, b). Although the electroactive CoFeOOH is intrinsically of poor conductivity, the nanoporous FeCo/$CeO_{2-x}N_x$ electrode has a low $R_I$ value of ~4.5 $\Omega$ (Supplementary Fig. 20c). This is probably due to the unique architecture of ultrathin CoFeOOH layer in-situ forming on the surface of interconnective conductive FeCo alloy skeleton, which is conducive to electron transportation during the OER processes.

Figure 3e compares the Tafel slope and current density of nanoporous FeCo/$CeO_{2-x}N_x$ at the overpotential of 300 mV with those of recently reported high-current-density OER electrocatalysts based on nonprecious metals (Supplementary Table 1)[14,41,45]. The nanoporous FeCo/$CeO_{2-x}N_x$ composite electrode has the lowest Tafel slope and the current density of as high as 1195 mA cm$^{-2}$ at overpotential of 300 mV, not only outperforming most nonprecious metal-based OER electrocatalysts but also satisfying the requirement of water electrolyzers for industrial application (>500 mA cm$^{-2}$ at overpotentials <300 mV)[11,12,44].

Figure 4a shows the Faradaic efficiency of nanoporous FeCo/$CeO_{2-x}N_x$ electrode for the OER, which is measured at a current density of ~1000 mA cm$^{-2}$. Therein, the Faradaic efficiency is determined by

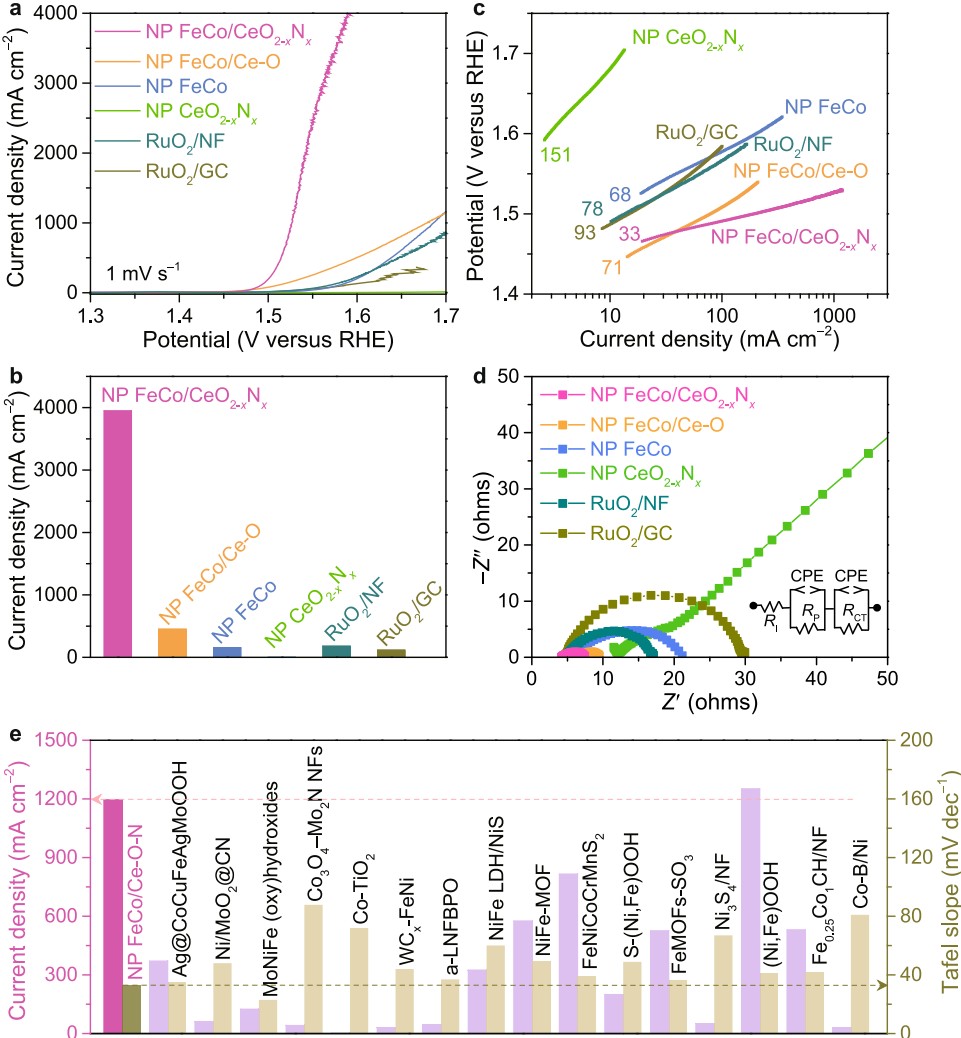

**Fig. 3 | Electrochemical properties of nanoporous electrodes. a** OER polarization curves for self-supported nanoporous (NP) FeCo/CeO$_{2-x}$N$_x$ (pink), FeCo/Ce-O (yellow), FeCo (blue), CeO$_{2-x}$N$_x$ (green) electrodes, RuO$_2$ electrodeposited on nickel foam (RuO$_2$/NF, dark green) and commercially available RuO$_2$ nanocatalyst immobilized on glassy carbon electrode (RuO$_2$/GC, dark yellow) in O$_2$-purged 1 M KOH electrolyte. Scan rate: 1 mV s$^{-1}$. **b** Comparison of current densities at the overpotential of 360 mV for self-supported NP FeCo/CeO$_{2-x}$N$_x$, FeCo/Ce-O, FeCo, CeO$_{2-x}$N$_x$, RuO$_2$/NF and RuO$_2$/GC electrodes. **c** Tafel plots comparing the Tafel slopes of different catalysts according to the OER polarization curves in panel (**a**).

**d** EIS spectra of NP FeCo/CeO$_{2-x}$N$_x$, FeCo/Ce-O, FeCo and CeO$_{2-x}$N$_x$ electrodes, as well as RuO$_2$/NF and RuO$_2$/GC. Inset: The electrical equivalent circuit used for fitting EIS, where $R_I$ and $R_{CT}$ denote the intrinsic electrode and electrolyte resistance and the charge transfer resistance, respectively, $R_P$ is the pore resistance, CPE represents the constant phase elements. **e** Comparison of current density at overpotential of 300 mV and Tafel slope for NP FeCo/CeO$_{2-x}$N$_x$ electrode with nonprecious metal-based OER catalysts previously reported (Supplementary Table S1).

comparing the experimental amount of gas generation with the theoretically calculated value. The coincidence of both values (near 100% Faradaic efficiency) indicates that no side reaction occurs on nanoporous FeCo/CeO$_{2-x}$N$_x$ during electrolysis after the formation of CoFeOOH/CeO$_{2-x}$N$_x$ heterostructure[28].

To investigate the OER durability of nanoporous FeCo/CeO$_{2-x}$N$_x$ electrode, the electrolysis measurement is performed at a potential of 1.54 V versus RHE in 1 M KOH at room temperature for over 400 h, where the current density always remains as high as ~1900 mA cm$^{-2}$ (Fig. 4b). Whereas a high current density usually accelerates the dissolution of metal ions due to a fast consumption rate of OH$^-$ and the localized pH change, there do not detect any Fe, Co, and Ce ions in the electrolyte at the different test time of 100, 200 and 400 h by inductively coupled plasma optical emission spectroscopy (ICP-OES) except for the trace amount of Fe (~0.0116 mg cm$^{-2}$) that is electrodeposited on the counter electrode of carbon rod (~16.5 cm$^{-2}$) after the stability test for 400 h. The fact reflects that the nanoporous FeCo/CeO$_{2-x}$N$_x$ also undergoes unavoidable dissolution of Fe during the rigorous OER

process, similar to other Fe-based electrocatalysts including mixed Ni-Fe and Co-Fe (oxy)hydroxides[67–72]. Nevertheless, the dissolution rate of Fe in nanoporous FeCo/CeO$_{2-x}$N$_x$ is evaluated to be ~0.477 μg h$^{-1}$ even at the current density of ~1900 mA cm$^{-2}$ (Supplementary Table 2). These observations are in sharp contrast with nanoporous FeCo/Ce-O and FeCo electrodes that encounter a remarkable reduction in current densities along with higher dissolution rates of Fe (~0.782 μg h$^{-1}$ and ~2.412 μg h$^{-1}$) in 100 h (Fig. 4b and Supplementary Table 2), reflecting the significant role of CoFeOOH/CeO$_{2-x}$N$_x$ heterostructure in improving the electrochemical stability of nanoporous FeCo/CeO$_{2-x}$N$_x$ electrode. This is probably due to evident electron transfer from CeO$_{2-x}$N$_x$ to CoFeOOH at their heterointerface (Supplementary Figure 13), which substantially stabilizes the chemical states of Fe component to alleviate the Fe leaching that is usually caused by the formation of soluble FeO$_4^{2-}$ under OER environment.[73] Specifically, the surface Fe component in nanoporous FeCo/CeO$_{2-x}$N$_x$ electrode has the Fe$^{3+}$/Fe$^{2+}$ ratio of 70.7: 29.3 after the durability test, the almost same as the initial value of 70.3: 29.7. As a consequence, it does not display

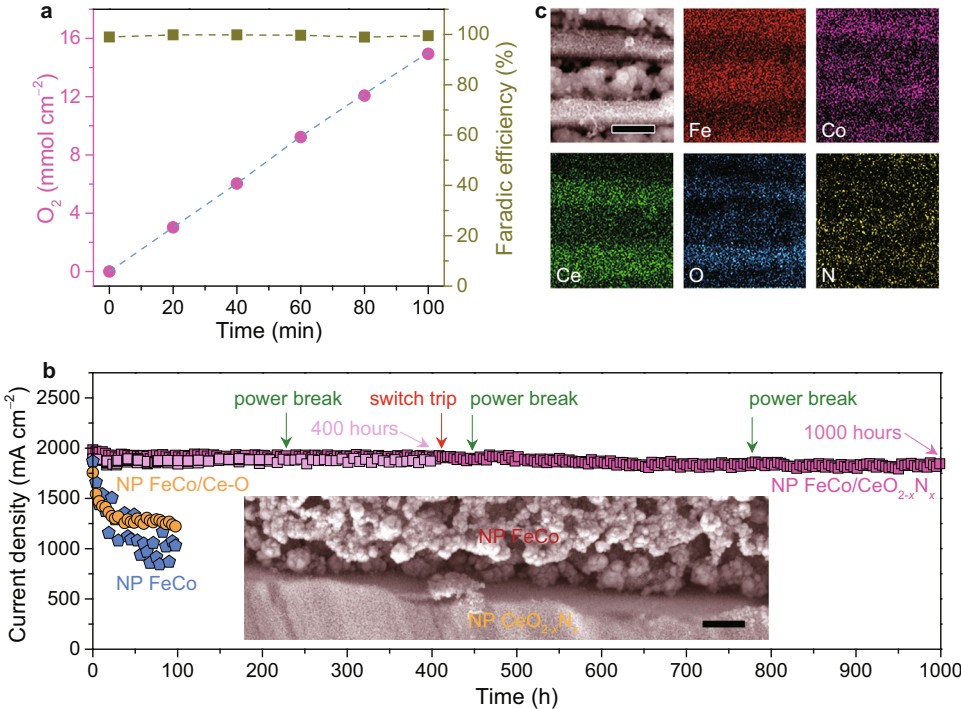

**Fig. 4 | OER performance of nanoporous FeCo/CeO$_{2-x}$N$_x$ electrode.**
**a** Comparison of oxygen volume (pink dots) and faradic efficiency (dark yellow squares) for nanoporous (NP) FeCo/CeO$_{2-x}$N$_x$ electrode and its theoretical value (dashed line) calculated based on the amount of consumed charges over the course of electrolysis. **b** Stability test (current density versus time) of NP FeCo/CeO$_{2-x}$N$_x$ electrode (pink squares) at 1.54 V versus RHE for 400 and 1000 h, respectively. During the 1000-hour stability test, there take place power break and switch trip accidents. Stability tests of NP FeCo/Ce-O (yellow circles) and NP FeCo (blue pentagons) electrodes were performed for 100 h at 1.81 and 1.76 V to deliver initial current densities that are comparable to the value of NP FeCo/CeO$_{2-x}$N$_x$ electrode. Inset: representative SEM image of NP FeCo/CeO$_{2-x}$N$_x$ electrode after performed at 1.54 V versus RHE for 1000 h. Scale bar, 500 nm. **c**, Typical SEM and EDS elemental mapping of Fe, Co, Ce, O, and N for NP FeCo/CeO$_{2-x}$N$_x$ electrode after performed at 1.54 V versus RHE for 1000 h. Scale bar, 500 nm.

evident changes in appearance and color after the durability test for 400 h (Supplementary Figure 21). Reproducibly, another nanoporous FeCo/CeO$_{2-x}$N$_x$ electrode also exhibits a stable current density of ~1900 mA cm$^{-2}$ even through some accidents such as power break and switch trip taking place during the durability test for 1000 h (Fig. 4b). Despite the nanoporous FeCo/CeO$_{2-x}$N$_x$ electrode encounters a slight composition evolution probably caused by the slow dissolution of Fe for 1000 h (Supplementary Figure 22), it still maintains initial laminate heterostructure of alternating nanoporous FeCo and CeO$_{2-x}$N$_x$ lamellas with sturdy interfaces (Fig. 4c and inset of Fig. 4b) and stable chemical states of surface Fe, Co and Ce atoms (Supplementary Figure 23). All these obversions demonstrate that the self-supported nanoporous FeCo/CeO$_{2-x}$N$_x$ electrodes exhibit exceptional long-term durability under very violent O$_2$ gas evolution, holding great promise as electrocatalytic materials of water oxidation reaction for large-scale energy storage of intermittently available renewable solar and wind sources.

## Discussion

In summary, we have developed hierarchical nanoporous alloy/oxy-nitride laminate composite electrodes by making use of symbiotic intermetallic compound lamellas as conformal templates in a facile and scalable alloying/dealloying and thermal nitridation procedure. These laminate composite electrodes are composed of periodically alternating nanoporous bimetallic iron-cobalt alloy and cerium oxy-nitride compound lamellas, wherein the former is prone to undergo surface reconstruction to form CoFeOOH and the latter maintains electrochemical stability under the strongly oxidative environment of OER. Owing to the CeO$_{2-x}$N$_x$ properly modulating the contiguous CoFeOOH to have near-optimal adsorption energies of *OH, *O and *OOH intermediates, the CoFeOOH/CeO$_{2-x}$N$_x$ heterostructure interfaces serve as the electroactive sites with remarkably enhanced activity. As a consequence of heterolamellas offering abundant electroactive sites of CoFeOOH/CeO$_{2-x}$N$_x$ interfaces and three-dimensional bicontinuous nanoporous architecture facilitating electron transfer and mass transport, the self-supported monolithic nanoporous FeCo/CeO$_{2-x}$N$_x$ composite electrodes exhibit superior alkaline OER electrocatalysis in 1 M KOH solution, with low onset overpotential (~186 mV) and Tafel slope (~33 mV dec$^{-1}$). The OER current density rapidly increases to ~1195 mA cm$^{-2}$ at the low overpotential of 300 mV. While extending the overpotential to 360 mV, they can reach an ultrahigh current density of >3900 mA cm$^{-2}$. Moreover, the nanoporous FeCo/CeO$_{2-x}$N$_x$ electrodes show exceptional stability for more than 1000 h even under very violent O$_2$ gas evolution at the current density of ~1900 mA cm$^{-2}$ and some accidents including power break and switch trip. These impressive electrochemical properties not only outperform commercially available RuO$_2$ and some representative OER catalysts recently reported but also enlist the nanoporous FeCo/CeO$_{2-x}$N$_x$ electrodes to hold great promise as an attractive OER catalyst for large-scale hydrogen generation via water electrolysis driven by intermittently available solar and wind power.

## Methods

### Preparation of lamella-heterostructured nanoporous FeCo/CeO$_{2-x}$N$_x$ electrode

Alloy ingots of Fe$_{25-y-z}$Co$_y$Ce$_z$Al$_{75}$ (at%, $y$, $z$ = 0, 5 or 25) were firstly produced by arc-melting pure Fe (99.98%), Co (99.99%), Ce (99.5%) and Al (99.95%) metals with different atomic ratios in an Ar atmosphere. Typically, molten Fe$_{15}$Co$_5$Ce$_5$Al$_{75}$ alloy was made by melting pure Fe, Co, Ce, and Al with an atomic ratio of 15: 5: 5: 75. After being cooled to room temperature in a water-assisted furnace, this alloy was cut into sheets with a thickness of ~400 μm and then chemically dealloyed to produce lamellar nanoporous FeCo/Ce-O composite

precursor in $N_2$-purged 6 M KOH aqueous solution at 85 °C until there do not observe any bubbles. After being rinsed in pure water (18.2 MΩ cm) to remove chemical substances in nanoporous channels and then dried in a vacuum environment, the nanoporous FeCo/Ce-O sheets were further thermally treated in a mixed atmosphere of argon/ammonia (Ar/NH₃) with the molar ratio of 90: 10 at 600 °C for 2 h to prepare self-supported lamellar and nanoporous FeCo/CeO$_{2-x}$N$_x$ heterostructure electrode materials. For comparison, the single-phase nanoporous FeCo alloy and nanoporous CeO$_{2-x}$N$_x$ individuals are fabricated by the same procedure on the basis of their corresponding alloy precursors, i.e., Fe$_{20}$Co$_5$Al$_{75}$ and Ce$_{25}$Al$_{75}$. For comparison, commercially available RuO$_2$ nanoparticles were casted on glassy carbon for electrochemical measurements. In addition, Ni foam supported RuO$_2$ nanoparticles were prepared by electrodeposition and calcination procedures, during which the CV electroplating was performed for 100 cycles within a potential window from −0.2 to 1.0 V at a scan rate of 50 mV s$^{-1}$ in 5 × 10$^{-3}$ M RuCl$_3$ electrolyte at 50 °C and then annealed at 200 °C for 2 h.

### Physicochemical characterizations

Microstructure characterizations and elements analysis of nanoporous electrodes were conducted on a thermal field emission scanning electron microscopy (JSM-7900F, JEOL, 5 kV) equipped with X-ray energy-dispersive spectroscopy. Low-magnification and high-resolution TEM images were obtained by a field-emission transmission electron microscope (JEOL JEM-2100F, 200 kV). X-ray diffraction measurements of nanoporous electrodes were performed on a Rigaku smartlab diffractometer with a monochromatic Cu Kα radiation. Chemical states of surface elements were analyzed using X-ray photoelectron spectroscopy (Thermo ECSALAB 250) with an Al anode. Charging effect was compensated by shifting binding energies according to the C 1s peak (284.8 eV). Raman spectra were collected on a micro-Raman spectrometer (Renishaw) equipped with a 532-nm-wavelength laser at a power of 0.5 mW. The concentrations of metal ions were measured by inductively coupled plasma optical emission spectroscopy (ICP-OES, Thermo electron).

### Electrochemical measurements

All electrochemical measurements were performed in a 1 M KOH aqueous solution based on a classic three-electrode setup, where self-supported nanoporous electrocatalysts (dimension: ~0.332 × ~0.253 × ~0.04 cm³, mass: ~9.25 mg) were directly used as the working electrode, with a carbon rod as the counter electrode and an Ag/AgCl within KCl-saturated solution as the reference electrode. Before the test, the reference electrode of Ag/AgCl reference electrode was calibrated by measuring the reversible hydrogen electrode (RHE) potential using a commercial Pt foil as the working electrode under a H₂-saturated 1 M KOH electrolyte. The potential is corrected according to the equation $E_{RHE} = E_{Ag/AgCl} + 1.016$ V and the iR-corrected one is determined in terms of the equation $E_{RHE} = E_{Ag/AgCl} + 1.016 - iR$. After the initial OER polarization test conducted in 1 M KOH electrolyte at a scan rate of 5 mV s$^{-1}$, the polarization curves of nanoporous electrodes were then collected at a scan rate of 1 mV s$^{-1}$ within a potential range from 1.3 V to 1.7 V (versus RHE) and their electrochemical impedance spectroscopy measurements were performed at the overpotential of 336 mV in frequency ranging from 10 mHz to 100 kHz. The electrochemical-specific areas were estimated by cyclic voltammogram measurements at various scan rates in a potential window of −0.35 to −0.25 V versus Ag/AgCl. According to the linear slope of scan rate versus current density, their double-layer capacitances ($C_{dl}$) were calculated according to the current density at −0.30 V versus Ag/AgCl against the scan rate. Electrochemical durability tests for nanoporous FeCo/CeO$_{2-x}$N$_x$, FeCo/Ce-O and FeCo electrodes were performed in 1 M KOH aqueous electrolyte at the potential of 1.54, 1.81, and 1.76 V (versus RHE), respectively.

### Density functional theory calculations

All the spin-polarized DFT calculations were performed by the Vienna ab initio simulation package (VASP) code with the method of projector augmented wave (PAW). The generalized gradient approximation (GGA) with Perdew-Burke-Ernzerhof (PBE) functional was utilized to describe the exchange-correlation potential. The Grimme method (DFT-D3 correction) was adopted to accurately describe the van der Waals interactions. As well, on-site Coulomb correction was adopted to correctly describe the electronic structures, and the effective U-J values were set to 5.3 and 3.3 eV for the 3$d$ state of Fe and Co, and a 5.0 eV U-J value was applied to the 4$f$ states of Ce. The kinetic cut-off energy was set to 400 eV, and the k-point sampling was set to 3 × 3 × 1 for integrating the Brillouin zones. We first established 2 × 4 × 1 supercell of CoFeOOH(001) and 3 × 2 × 2 one of CeO$_2$(111) after their ($\sqrt{3}$ × 1) R30° reconstructions with the lattice angles adjusting to 90°. Therein, the dimensions of CoFeOOH(001) and CeO$_2$(111) supercells are 12.97 Å × 11.24 Å and 13.25 Å × 11.48 Å, respectively. To construct the amorphous structure, ab initio molecular dynamics (AIMD) calculations were performed on CoFeOOH(001) slab under the NVT condition at 300 K, based on the experimental condition. The total simulation time was 5 ps with a time step of 1 fs (5000 simulation steps)[74]. After the formation of amorphous CoFeOOH, we further constructed atomic models of CoFeOOH/CeO$_{2-x}$N$_x$, CoFeOOH/CeO$_2$ heterostructures by combining the amorphous CoFeOOH layer and the N-doped or undoped Ce terminal CeO$_2$(111) slab. The vacuum region of 15 Å was set to avoid the interaction between periodic images. Notably, each step in these model constructions was accompanied by a geometry optimization calculation, wherein the convergence criterions of the energy and force were set to 1 × 10$^{-5}$ eV and 0.02 eV Å$^{-1}$, respectively. The Gibbs free energy change ($\Delta G$) is calculated by: $\Delta G = \Delta E + \Delta ZPE - T\Delta S$, where $\Delta E$ is the reaction energy, $\Delta ZPE$ is the change of zero-point energy, $T$ is the temperature (298.15 K) and $\Delta S$ is the change of entropy, respectively.

### Reporting summary

Further information on research design is available in the Nature Portfolio Reporting Summary linked to this article.

## Data availability

All data supporting this study and its findings within the article and its Supplementary Information are available from the corresponding author [Xing-You Lang] upon reasonable request.

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

## Acknowledgements
This work was supported by the National Natural Science Foundation of China (No. 52271217 (X.Y.L.), 52201217 (H.S.), 52130101 (Q.J.), 51871107 (X.Y.L.)), China Postdoctoral Support Program for Innovation Talents (BX20220129 (H.S.)), China Postdoctoral Science Foundation (2022M711290 (H.S.)), Chang Jiang Scholar Program of China (Q2016064 (X.Y.L.)), the Program for JLU Science and Technology Innovative Research Team (JLUSTIRT, 2017TD-09 (Q.J.)), the Fundamental Research Funds for the Central Universities, and the Program for Innovative Research Team (in Science and Technology) in University of Jilin Province.

## Author contributions
X.Y.L. and Q.J. conceived and designed the experiments. S.P.Z., H.S., Y.L., Z.W., G.F.H., W.Z., X.Y.L,. and W.T.Z. carried out the fabrication of materials, and performed the electrochemical measurements and microstructural characterizations. T.Y.D., T.H.W., and Q.J. performed DFT simulation. X.Y.L. and Q.J. wrote the paper, and all authors discussed the results and commented on the manuscript.

## Competing interests
The authors declare no competing interests.
