## [Peer Review File · Nature Communications]

REVIEWER COMMENTS

Reviewer #1 (Remarks to the Author):

The authors aimed to address the stability and activity of water oxidation electrodes at high current density. They developed hierarchical nanoporous alloy/oxy-nitride laminate composite electrodes for oxygen evolution reaction. They form an Al₁₃(FeCo)₄ - Al₁₁Ce₃ eutectic and make the FeCo-active phase porous by dealloying of Al, followed by nitridation. A lamellar electrode can perform at high current density and low overpotential. While the use of FeCoOOH catalyst in the composite form might be interesting from a technological point of view, the manuscript does not explain any new approaches towards stabilization of the electrodes for OER and is not suitable for publication in Nature Communications.

I would like to express a few major concerns:

-The authors claim to have prepared the "self-supported hybrid electrodes composed of alternating nanoporous bimetallic iron-cobalt alloy and cerium oxy-nitride (FeCo/Ce-O-N) heterolamellas". I would like to note that the term hybrid material is commonly reserved for organic-inorganic composites, thus in this case it should be avoided. Also, the electrode is not a bimetallic alloy. The alloy serves as a precursor towards oxyhydroxide catalyst, which is evident from CV and XPS data.

-Furthermore, the authors claim that the active phase is 'Co-doped Fe₃O₄, or (FeCo)₃O₄', I suggest they think about the concept of doping.

-SEM analysis is of poor quality and analyzes small areas, it is not clear if the deformations between lamellas are from the sample processing or de-alloying. I suggest authors polish the cross-sections before the analysis.

-In general, it is not clear to me what is the amount of material at the electrode, the amount of material can drastically affect the stability and 'activity' when normalized to the geometric surface area.

- It is claimed that the material has high intrinsic activity and stability due to the interaction of CoFeOOH with Ce-O-N ('Owing to the presence of highly electroactive FeCoOOH/Ce-O-N interfaces'). However, the number of interfaces compared to bulk CoFe-phase is minor, and cannot drastically affect the overall activity, thus I see no need for DFT studies of those interfaces.

- It is unfair to compare the electrode performance towards 'commercially available RuO₂ nanocatalyst immobilized on glassy carbon electrode by dint of Nafion as polymer binder'. There is a mature technology called dimensionally stable anodes, that serves as a state-of-the-art for practical application.

-EIS analysis is unclear, at what point was it done, and after how many cycles, was the electrode already in FeCoOOH form, if not, it is possible that EIS was measured for corrosion and not for OER.

-My biggest concern is related to stability. While the current density is stable, we have to note that it was shown to be a poor indicator of stability. The authors should analyze the morphology of the electrode after the test, they should analyze the Fe content in the electrolyte at some points during the stability test and at the end. The Fe and Co content at the electrode should be analyzed before and after the test and the deposition of metals on the counter (HER) electrode should be considered. I suggest the authors study the papers dealing with the stability of Ni and NiFe systems (10.1021/acsaem.1c02604, 10.1002/chem.201803165 , 10.1038/s41929-020-0496-z, 10.1021/acscatal.0c01813, 10.1016/j.chempr.2017.03.006).

Reviewer #2 (Remarks to the Author):

This work is entitled "Lamella-heterostructured nanoporous FeCo/Ce-O-N electrodes as ultrahigh-current-density and stable catalysts for oxygen evolution". This work reported a rare way, a combination of alloying/dealloying and basic etching of Al, to make highly porous, active electrocatalysts for OER. This method is indeed interesting, and the lamella structure may be a good lead to another round of OER catalyst design. Of course, the activities would've been even higher too. However, there are some critical concerns about the major claim of this work. Prior to the acceptance, the reviewer would recommend a major revision of this work. Please see the comments below:

1. One of the major activity contributors is Ce-O-N. According to the XRD indexing, it seems the better way to present this species would be $\text{CeO}_{2-z}\text{N}_{2/3z}$?
2. Please reason why the starting composition is $\text{Fe}_{25-x-y}\text{Co}_x\text{Ce}_y\text{Al}_{75}$ (at%, where x, y = 0, 5 or 25)?
3. Be careful about your deconvolution of $\text{Ce}^{3+/4+}$. It does not seem the deconvolution was done correctly; for example, do the authors use the identical full width at half maximum to analyze the data? There are satellite peaks in 890-880 eV [Physical Chemistry Chemical Physics 2017, 19, 31545-31552.]
4. According to the authors, this Ce-O-N species can be stably existing in a highly basic OER condition. Yet it is also well known in the literature and across the field that all the metal nitrides, phosphides, sulfides..etc. are turning themselves into oxides after an aggressive OER. The characterization to the stability of Co-O-N seems only between the first-second scan. Not convincing and fail to support the title, also the main message, of the entire work.
5. In the microstructural characterization, SEM of the as-prepared sample would be important. Since all the SEM images of the CoFe/Co-O-N were acquired after the tests. It's hard to compare the before and after, and thus fail to support the point of durability.

6. In Fig. 1b, the NPFeCo looks like a film at the interfaces, rather than a particulate substance. In addition, the FFT patterns in e and f are identical information from the hr images. No new confirmation was done. Since the materials are in micron scale, the SAED of the composite alloy should be easy to acquire, for the purpose of structural reconfirmation.
7. In page 9, starting line 172. How come all the data show metal cations ($\text{Co}^{2+}/3+$, $\text{Fe}^{3+}/2+$)? The NP CoFe shouldn't be metallic?
8. The XPS in both Fig 1 and 2 should be plotted together, since the authors used them to compare the changes before and after the OER tests.
9. It is also very well known that real active sites for OER is amorphous hydroxide/oxyhydroxide. The entire theoretical calculation part is based on crystalline models. How do the authors reason themselves for using crystalline calculations to support the amorphous activities?
10. The benchmark of OER performance is RuO₂ on a glassy carbon electrode with organic binder. Yet the working electrode for CoFe/Co-O-N is a self-supported manner. This comparison was not fairly done due to the huge contact resistance for the RuO₂ case. The authors should alloy the RuO₂ with a conductive electrode (e.g. Ni foam) at high temperature to give a fair comparison. Of course, no Nafion binder should be used to block the active sites of RuO₂.
11. The impedance of Fig. 3d is confusing. How come the mass transport parts for all the critical samples are missing? This is important because the ECSA seems to be a critical impact on the OER activity (56 times higher).
12. It is really amazing that using a carbon rod as the HER electrode can yield such a great OER current. The reviewer would say the LSV data provided by the authors are even much better than using Pt as a HER electrode in many other works in the field. It is known that carbon rod is much more inert, and not even have a chance to compare with Pt. The geometry dimension (width, length, and height) of the self-support electrode of CoFe/Co-O-N should be provided to answer the concern regarding such the high current density.
13. In Fig. 4, the stability tests. Please use elemental leaching in the electrolyte to support the argument for the high stability of OER at high currents. The EDS only tells you how many much the quantity of active elements remains there before leaching out. In addition, please also provide the appearance change of the working electrode before and after the 1000 hour test. Shall there be a significant color change of the electrocatalyst, provide an explanation about that.

Reviewer #3 (Remarks to the Author):

This work by Zeng et al. reported a laminate composite electrode composed of alternating hierarchical nanoporous bimetallic iron-cobalt alloy/cerium oxynitride (FeCo/Ce-O-N) heterolamellas as self-supported electrocatalytic material for oxygen evolution reaction. Owing to the unique architecture that offer abundant electroactive sites and facilitate electron transfer and ion transportation, the nanoporous FeCo/Ce-O-N electrode delivers ultrahigh current densities at low overpotentials and retains exceptional stability at high current densities for more than 1000 hours. This electrode is an attractive OER catalyst and holds a promise for practical application in large-scale hydrogen production via electrochemical water-splitting technologies. The manuscript is well written with clear logic. This work is novel and important to the field. I would recommend its publication in Nature Communications after the authors addressing the following points:

1. In Figure 1h, the authors should explain why there observe H₂O and HO in the XPS spectrum of O 1s for the as-annealed nanoporous FeCo/Ce-O-N specimen.
2. In view that the authors mention that the current density of nanoporous FeCo/Ce-O-N can reach as high as >3900 mA cm⁻², they should extend the current density in Figure 3a.
3. The authors should perform supplementary experiment to demonstrate the reproducibility of FeCo/Ce-O-N hybrid electrode.
4. The authors should provide EDS spectra of precursor alloys to demonstrate their atomic ratios of Fe, Co, Ce, Al.
5. Some recently published related articles are suggested to be cited, for examples, Nature communications, 2022, 13, 2191; Nature communications, 2020, 11, 1664 ; Electrochem. Energy Rev. 2021, 4(1), 136–145; Nano-Micro Letters, 2022, 14, 120; Electrochem. Energy Rev. 2021, 4(3), 566–600; SusMat, 2021, 1(4): 460-481; Chem. Soc. Rev., 2022,51, 4583-4762; etc.

Response To Reviewers' Comments

Reviewer #1 (Remarks to the Author):

The authors aimed to address the stability and activity of water oxidation electrodes at high current density. They developed hierarchical nanoporous alloy/oxynitride laminate composite electrodes for oxygen evolution reaction. They form an $Al_{13}(FeCo)_4-Al_{11}Ce_3$ eutectic and make the FeCo-active phase porous by dealloying of Al, followed by nitridation. A lamellar electrode can perform at high current density and low overpotential. While the use of FeCoOOH catalyst in the composite form might be interesting from a technological point of view, the manuscript does not explain any new approaches towards stabilization of the electrodes for OER and is not suitable for publication in Nature Communications. I would like to express a few major concerns:

Reply: We thank the reviewer for finding potential interest of our work. We also appreciate the reviewer for his/her insightful comments and constructive suggestions. Following these comments/suggestions, we have supplemented some experiments, including SEM characterization on nanoporous FeCo/CeO_{2-x}N_x, ICP-OES analysis of electrolytes during the long-term durability test of nanoporous FeCo/CeO_{2-x}N_x at 1.54 V, ICP-OER analysis of electrodeposited metals on the counter electrode of carbon rod after the durability test, OER measurements on RuO₂-based electrodes (RuO₂/NF that is prepared by electrodepositing RuO₂ on nickel foam and commercially available Ti mesh-supported RuO₂ DSA). Based on these results, we have systemically discussed the approaches towards stabilization of electrodes for the OER, i.e., the hierarchical nanoporous architecture to accelerate mass transportation of OH⁻ and thus keep a stable environment of electroactive site, which substantially lowers the unavoidable dissolution of Fe ions particularly at high current density of ~1900 mA cm⁻², and the laminate heterostructure of alternating nanoporous FeCo and CeO_{2-x}N_x lamellas with steady interfaces, which support each other to withstand violent gas evolution during the long-term durability test. In addition, we have completely revised the manuscript following other constructive comments and suggestions. The detailed corrections are listed below. In view that the laminate composite electrode with a novel architecture of alternating nanoporous FeCo and CeO_{2-x}N_x lamellas exhibit exceptionally high electrocatalytic activity and ultralong stability for the OER, as pointed out by all Reviewers, "*the use of FeCoOOH catalyst in the composite form might be interesting from a technological point of view*" (the comment of this Reviewer), "*This method is indeed interesting, and the lamella structure may be a good lead to another round of OER catalyst design*" (Reviewer #2), and "*This electrode is an attractive OER catalyst and holds a promise for practical application*

*in large-scale hydrogen production" (Reviewer #3), we wish the reviewer could share our confidence and belief that the work reported in this paper deserves to be published in a high-impact journal, like *Nature Communications*.*

(1) The authors claim to have prepared the "self-supported hybrid electrodes composed of alternating nanoporous bimetallic iron-cobalt alloy and cerium oxynitride (FeCo/Ce-O-N) heterolamellas". I would like to note that the term hybrid material is commonly reserved for organic-inorganic composites, thus in this case it should be avoided. Also, the electrode is not a bimetallic alloy. The alloy serves as a precursor towards oxyhydroxide catalyst, which is evident from CV and XPS data.

Reply: We appreciate the reviewer for the constructive suggestions. According to these suggestions, we have corrected these presentations in the revised manuscript.

(2) Furthermore, the authors claim that the active phase is 'Co-doped Fe₃O₄, or (FeCo)₃O₄', I suggest they think about the concept of doping.

Reply: Following this suggestion, we have corrected "Co doped Fe₂O₃, or (FeCo)₂O₃", "Co doped Fe₃O₄, or (FeCo)₃O₄" as "Co incorporated Fe₂O₃, or Co-Fe₂O₃" and "Co-Fe₃O₄", respectively.

(3) SEM analysis is of poor quality and analyzes small areas, it is not clear if the deformations between lamellas are from the sample processing or de-alloying. I suggest authors polish the cross-sections before the analysis.

Reply: We thank the reviewer for the constructive comment and suggestion. According to this suggestion, we have reperformed SEM characterization of nanoporous FeCo/CeO_{2-x}N_x laminate composite electrode. The representative SEM image has been renewed in **Figure 1b** and Supplementary **Figure 4** with large areas, displaying the unique architecture consisting of periodically alternating nanoporous FeCo alloy and CeO_{2-x}N_x lamellas.

(4) In general, it is not clear to me what is the amount of material at the electrode, the amount of material can drastically affect the stability and 'activity' when normalized to the geometric surface area.

Reply: We thank the reviewer for this insightful comment, according to which we have provided the amount of materials in Method. The geometric dimension of tested materials is $\sim 0.332 \times \sim 0.253 \times \sim 0.04 \text{ cm}^3$, while their mass is $\sim 9.25 \text{ mg}$.

(5) It is claimed that the material has high intrinsic activity and stability due to the

interaction of CoFeOOH with Ce-O-N (‘Owing to the presence of highly electroactive FeCoOOH/Ce-O-N interfaces’). However, the number of interfaces compared to bulk CoFe-phase is minor, and cannot drastically affect the overall activity, thus I see no need for DFT studies of those interfaces.

Reply: We appreciate the reviewer for the insightful comment. We agree with the reviewer that the number of interfaces of FeCo/CeO_{2-x}N_x indeed is minor compared with the constituent nanoporous FeCo lamellas. Nevertheless, the unique architecture of alternating nanoporous FeCo and CeO_{2-x}N_x lamellas enlists nanoporous FeCo/CeO_{2-x}N_x electrode to exhibit higher electrocatalytic activity and durability than nanoporous FeCo. Despite the electrochemical surface area (ECSA) of nanoporous FeCo/CeO_{2-x}N_x electrode is estimated to be ~1.6 fold of that of nanoporous FeCo, it does not account for ~25-fold increment in geometric area-normalized current density. This implies the significant role of FeCoOOH/CeO_{2-x}N_x interface with exceptional intrinsic activity towards for the OER. This is also verified by DFT simulations on amorphous FeCoOOH and FeCoOOH/CeO_{2-x}N_x heterostructure. As shown in **Figure 2d**, the OER suffers from the potential-determined step (PDS) of O-O bond formation on the CoFeOOH/CeO_{2-x}N_x, which mediates O* to react with another OH⁻ to form the HOO* intermediate. This is different from the bare CoFeOOH, which undergoes the PDS of the deprotonation of *OH for the formation of *O. Owing to the influence of Ce-N bonds, the Co atoms at the CoFeOOH/CeO_{2-x}N_x interface have lower adsorption energies of *OH, *O and *OOH intermediates, giving rise to the ΔG_{PDS} value of as low as ~0.44 eV, in sharp contrast with that of amorphous CoFeOOH (~0.87 eV). The low energy barrier enlists the CoFeOOH/CeO_{2-x}N_x interface as the electroactive sites to substantially boost OER kinetics.

(6) It is unfair to compare the electrode performance towards ‘commercially available RuO₂ nanocatalyst immobilized on glassy carbon electrode by dint of Nafion as polymer binder’. There is a mature technology called dimensionally stable anodes, that serves as a state-of-the-art for practical application.

Reply: We thank the reviewer for the constructive suggestion. According to this suggestion, we have purchased one dimensionally stable anode, i.e., Ti mesh supported RuO₂ nanoparticles (RuO₂ DSA), and measured its OER electrocatalytic properties. The detailed result has been shown in Supplementary **Figure 13**. Compared with the commercially available RuO₂ DSA electrode, our nanoporous FeCo/CeO_{2-x}N_x electrode exhibits a superior electrocatalytic behavior.

(7) EIS analysis is unclear, at what point was it done, and after how many cycles, was the electrode already in FeCoOOH form, if not, it is possible that EIS was measured

for corrosion and not for OER.

Reply: According to this comment, we have provided the detailed information on EIS measurement in Method section. The EIS analysis is conducted on the nanoporous FeCo/CeO_{2-x}N_x electrode after the initial OER test, where the FeCoOOH has already formed. Therefore, the EIS analysis reflects the properties for the OER not for the corrosion.

(8) My biggest concern is related to stability. While the current density is stable, we have to note that it was shown to be a poor indicator of stability. The authors should analyze the morphology of the electrode after the test, they should analyze the Fe content in the electrolyte at some points during the stability test and at the end. The Fe and Co content at the electrode should be analyzed before and after the test and the deposition of metals on the counter (HER) electrode should be considered. I suggest the authors study the papers dealing with the stability of Ni and NiFe systems (10.1021/acsaem.1c02604, 10.1002/chem.201803165, 10.1038/s41929-020-0496-z, 10.1021/acscatal.0c01813, 10.1016/j.chempr.2017.03.006).

Reply: We thank the reviewer for the insightful comment and constructive suggestion. Following this suggestion and the methods reported in these papers, we have additionally performed electrochemical stability test of nanoporous FeCo/CeO_{2-x}N_x electrode for 400 hours at the current density of ~1900 mA cm⁻², during which we have analyzed the Fe, Co and Ce contents in the electrolytes using ICP-OES after performing for 100, 200, 300 and 400 hours, respectively. The concentrations of Fe, Co and Ce ions in the tested electrolyte are too low to be detected by ICP-OES. In addition, we also have conducted additional ICP-OES analysis for the deposited metals on the counter electrode of carbon rod. According to ICP-OES measurement, there is only Fe to electrodeposit on the carbon rod and the specific mass of Fe is determined to be only ~0.0116 mg cm⁻² during the OER durability test for 400 hours, according to which the dissolution rate of Fe in nanoporous FeCo/CeO_{2-x}N_x is evaluated to be 0.477 μg h⁻¹. The dissolution rate is lower than other Fe-based electrocatalysts for the OER particularly at high current density. This is probably due to the hierarchical nanoporous structure that ensures fast transportation of OH⁻ to keep a stable environment at active sites, lowering the dissolution rate of metal ions. This accounts for the slight composition evolution of nanoporous FeCo/CeO_{2-x}N_x electrode after the durability test for 1000 hours (Supplementary **Figure 18**). Nevertheless, it still maintains initial laminate heterostructure of alternating nanoporous FeCo and CeO_{2-x}N_x lamellas with sturdy interfaces (**Figure 4c** and inset of **Figure 4b**) and stable chemical states of surface Fe, Co and Ce atoms (Supplementary **Figure 19**) due to its exceptional steady heterostructure. As a result,

nanoporous FeCo/ CeO_{2-x}N_x electrode exhibits the stable current density of ~1900 mA cm⁻² even through some accidents such as power break and switch trip take place during the durability test for 1000 hours (**Figure 4b**). These results indicate the outstanding durability of nanoporous FeCo/CeO_{2-x}N_x electrode under very violent O₂ gas evolution, holding great promise as electrocatalytic materials of water oxidation reaction for large-scale energy storage of intermittently available renewable solar and wind sources. Besides, these papers have been listed in Reference.

Reviewer #2 (Remarks to the Author):

This work is entitled "Lamella-heterostructured nanoporous FeCo/Ce-O-N electrodes as ultrahigh-current-density and stable catalysts for oxygen evolution". This work reported a rare way, a combination of alloying/dealloying and basic etching of Al, to make highly porous, active electrocatalysts for OER. This method is indeed interesting, and the lamella structure may be a good lead to another round of OER catalyst design. Of course, the activities would've been even higher too. However, there are some critical concerns about the major claim of this work. Prior to the acceptance, the reviewer would recommend a major revision of this work. Please see the comments below:

Reply: We thank the reviewer for finding potential interest of our work and recommending acceptance for publication in *Nature Communications*. We also appreciate the reviewer for his/her insightful comments and constructive suggestions. Following these comments/suggestions, we have supplemented some experiments, including SAED characterization of nanoporous FeCo/CeO_{2-x}N_x electrode, optical characterization of nanoporous FeCo/CeO_{2-x}N_x electrode after the durability test, ICP-OES analysis of Fe, Co and Ce ions in electrolyte during the durability test of nanoporous FeCo/CeO_{2-x}N_x, ICP-OES analysis of metals deposited on the counter electrode of carbon rod after the durability test, DFT simulations on amorphous FeCoOOH with/without CeO_{2-x}N_x, electrochemical measurement of OER based on RuO₂/NF that is prepared by electrodepositing RuO₂ on nickel foam. Based on these results, we have completely revised the manuscript. The detailed corrections are listed below.

(1) One of the major activity contributors is Ce-O-N. According to the XRD indexing, it seems the better way to present this species would be CeO_{2-z}N_{2/3z}?

Reply: We thank the reviewer for the constructive suggestion. Following this suggestion, we have modified the presentation of Ce-O-N species as CeO_{2-x}N_x in the revised manuscript.

(2) Please reason why the starting composition is Fe_{25-x-y}Co_xCe_yAl₇₅ (at%, where x, y = 0, 5 or 25)?

Reply: We appreciate the reviewer for the constructive suggestion. The reason why we choose the starting composition of Fe_{25-y-z}Co_yCe_zAl₇₅ (at%, where y, z = 0, 5 or 25) is to ensure there form single- or multi-phase nanostructures, by which the compositions and architectures of nanoporous electrodes are adjusted during the chemical dealloying process. For y = 5 and z = 0, i.e., Fe₂₀Co₅Al₇₅, there mainly forms

single-phase $\text{Al}_{13}(\text{FeCo})_4$ intermetallic compound; For $y = 0$ and $z = 25$, i.e., $\text{Ce}_{25}\text{Al}_{75}$, there produces almost single-phase $\text{Al}_{11}\text{Ce}_3$ intermetallic compound. When adding Co and Ce with $y = z = 5$, there form multi-phase nanostructure of alternating intermetallic $\text{Al}_{13}(\text{FeCo})_4$ and $\text{Al}_{11}\text{Ce}_3$ lamellas. Based on these different precursor alloys, three nanoporous electrodes, i.e., nanoporous $\text{FeCo}/\text{CeO}_{2-x}\text{N}_x$ laminate composite, and nanoporous FeCo and $\text{CeO}_{2-x}\text{N}_x$ individuals, are prepared by chemical dealloying and thermal nitridation processes.

(3) Be careful about your deconvolution of $\text{Ce}^{3+}/^{4+}$. It does not seem the deconvolution was done correctly; for example, do the authors use the identical full width at half maximum to analyze the data? There are satellite peaks in 890-880 eV [Physical Chemistry Chemical Physics 2017, 19, 31545-31552.]

Reply: We appreciate the reviewer for the insightful comment. According to this comment, we have reanalyzed XPS spectra of Ce 3d with the identical full width at half maximum based on the references including Phys. Chem. Chem. Phys. 19 (2017) 31545-31552, Nat. Catal. 2 (2019) 334-341 and Nat. Commun. 11 (2020) 4240. The corrected Ce 3d XPS spectra have been shown in **Figure 1h**, **Figure 2c**, Supplementary **Figure 6** and Supplementary **Figure 14**. According to previous reports on Ce 3d XPS spectra in Phys. Chem. Chem. Phys. 19 (2017) 31545-31552, Nat. Catal. 2 (2019) 334-341 and Nat. Commun. 11 (2020) 4240, there do not observe satellite peaks in 890-880 eV except for the ones corresponding to Ce $3d_{5/2}$ of Ce^{3+} and Ce^{4+} . The literature has been listed in the References.

(4) According to the authors, this Ce-O-N species can be stably existing in a highly basic OER condition. Yet it is also well known in the literature and across the field that all the metal nitrides, phosphides, sulfides..etc. are turning themselves into oxides after an aggressive OER. The characterization to the stability of Ce-O-N seems only between the first-second scan. Not convincing and fail to support the title, also the main message, of the entire work.

Reply: We thank the reviewer for the comment. We agree with the reviewer that metal nitrides, phosphides, sulfides usually undergo surface reconstruction into oxides during the aggressive OER. Different from those metal nitrides, the constituent $\text{CeO}_{2-x}\text{N}_x$ in nanoporous $\text{FeCo}/\text{CeO}_{2-x}\text{N}_x$ laminate composite electrode keeps exceptionally electrochemical stability during the OER. To demonstrate this fact, we have additionally conducted XPS analysis of nanoporous $\text{FeCo}/\text{CeO}_{2-x}\text{N}_x$ laminate composite electrode after performing six OER polarization curves. The detailed results are shown in **Figure 2c** for high-resolution XPS of Ce $3d_{5/2}$, Co $2p_{3/2}$, Fe $2p_{3/2}$, O 1s and N 1s. As shown in **Figure 2c**, the surface Ce atoms keep the almost constant

$\text{Ce}^{4+}/\text{Ce}^{3+}$ ratio of 75/25 compared with the ones in the pristine nanoporous $\text{FeCo}/\text{CeO}_{2-x}\text{N}_x$ electrode (**Figure 1h**). This observation demonstrates the exceptional stability of the constituent $\text{CeO}_{2-x}\text{N}_x$ probably due to the N doping in CeO_2 , not the formation of cerium nitride. Considering this fact, we have fully corrected the presentation of “Ce-O-N” as “ $\text{CeO}_{2-x}\text{N}_x$ ” in the revised manuscript.

(5) In the microstructural characterization, SEM of the as-prepared sample would be important. Since all the SEM images of the CoFe/Co-O-N were acquired after the tests. It's hard to compare the before and after, and thus fail to support the point of durability.

Reply: We thank the reviewer for this comment. Actually, **Figure 1b** and Supplementary **Figure 4** show a representative SEM image of as-prepared nanoporous $\text{FeCo}/\text{CeO}_{2-x}\text{N}_x$ electrode. Differently, Supplementary **Figure 8** shows the SEM image and its corresponding EDS elemental mapping images of nanoporous $\text{FeCo}/\text{CeO}_{2-x}\text{N}_x$ after initial OER test, where the constituent nanoporous FeCo lamellas undergo surface oxidation in aggressive OER while the $\text{CeO}_{2-x}\text{N}_x$ ones maintain electrochemical stability. In contrast, inset of **Figure 4b** and **Figure 4c** show typical SEM images of nanoporous $\text{FeCo}/\text{CeO}_{2-x}\text{N}_x$ after the long-term stability measurement for 1000 hours. As shown in these typical SEM images, the nanoporous $\text{FeCo}/\text{CeO}_{2-x}\text{N}_x$ electrode always exhibits the unique architecture consisting of periodically alternating nanoporous FeCo alloy/oxide and $\text{CeO}_{2-x}\text{N}_x$ lamellas no matter in initial OER test or long-term stability measurement, demonstrating the outstanding structural durability. In addition, we have additionally performed electrochemical stability test of nanoporous $\text{FeCo}/\text{CeO}_{2-x}\text{N}_x$ electrode for 400 hours at the current density of $\sim 1900 \text{ mA cm}^{-2}$, during which we have analyzed the Fe, Co and Ce contents in the electrolytes using ICP-OES after performing for 100, 200, 300 and 400 hours, respectively. The concentrations of Fe, Co and Ce ions in the tested electrolyte are too low to be detected by ICP-OES. In addition, we also have conducted additional ICP-OES analysis for the deposited metals on the counter electrode of carbon rod. According to ICP-OES measurement, there is only Fe to electrodeposit on the carbon rod and the specific mass of Fe is determined to be only $\sim 0.0116 \text{ mg cm}^{-2}$ during the OER durability test for 400 hours, according to which the dissolution rate of Fe in nanoporous $\text{FeCo}/\text{CeO}_{2-x}\text{N}_x$ is evaluated to be $0.477 \mu\text{g h}^{-1}$. This accounts for the composition evolution of nanoporous $\text{FeCo}/\text{CeO}_{2-x}\text{N}_x$ electrode after the durability test for 1000 hours (Supplementary **Figure 18**). Nevertheless, it still maintains initial laminate heterostructure of alternating nanoporous FeCo and $\text{CeO}_{2-x}\text{N}_x$ lamellas with sturdy interfaces (**Figure 4c** and inset of **Figure 4b**) and stable chemical states of surface Fe, Co and Ce atoms (Supplementary **Figure 19**) due

to its exceptional steady heterostructure. As a result, nanoporous FeCo/ CeO_{2-x}N_x electrode exhibits the stable current density of ~1900 mA cm⁻² even through some accidents such as power break and switch trip take place during the durability test for 1000 hours (**Figure 4b**). These results indicate the outstanding durability of nanoporous FeCo/CeO_{2-x}N_x electrode under very violent O₂ gas evolution.

(6) In Fig. 1b, the NP FeCo looks like a film at the interfaces, rather than a particulate substance. In addition, the FFT patterns in e and f are identical information from the hr images. No new confirmation was done. Since the materials are in micron scale, the SAED of the composite alloy should be easy to acquire, for the purpose of structural reconfirmation.

Reply: We thank the reviewer for the comment and constructive suggestion. The as-prepared nanoporous FeCo/CeO_{2-x}N_x composite electrode is composed of alternating nanoporous FeCo alloy/oxide lamellas and nanoporous CeO_{2-x}N_x lamellas. Therefore, it is reasonable that the nanoporous FeCo lamellas look like films at the interfaces. In addition, we also have carried out supplementary SAED characterization according to the constructive suggestion. The detailed result has been shown in inset of **Figure 1d**, where different lattice planes of FeCo and CeO_{2-x}N_x can be identified in the diffraction rings. The planes (111), (200), (220) and (311) have been assigned to the face-centered cubic (fcc) phase of CeO_{2-x}N_x, and (110) and (200) to the body-centered cubic (bcc) phase of FeCo, respectively, on the basis of interplanar distance.

(7) In page 9, starting line 172. How come all the data show metal cations (Co²⁺/³⁺, Fe³⁺/²⁺)? The NP CoFe shouldn't be metallic?

Reply: We thank the reviewer for the comment. According to this comment, we have double-checked the XPS spectra of Co 2p and Fe 2p of as-prepared nanoporous FeCo/CeO_{2-x}N_x electrode. As shown in **Figure 1h**, the surface Co and Fe atoms indeed present mainly in oxidized states due to the formation of thick surface Co-Fe₃O₄ layer, which will facilitate the conversion into CoFeOOH under the initial OER test.

8. The XPS in both Fig 1 and 2 should be plotted together, since the authors used them to compare the changes before and after the OER tests.

Reply: We thank the reviewer for the suggestion. Following this suggestion, we have plotted high-resolution XPS spectra of nanoporous FeCo/CeO_{2-x}N_x electrode before and after initial OER test in Supplementary **Figure 9** for comparison.

9. It is also very well known that real active sites for OER is amorphous hydroxide/oxyhydroxide. The entire theoretical calculation part is based on crystalline models. How do the authors reason themselves for using crystalline calculations to support the amorphous activities?

Reply: According to this comment, we have performed DFT simulations on amorphous FeCoOOH with/without CeO_{2-x}N_x. The detailed results have been shown in **Figure 2d,e**. Evidently, the OER suffers from the potential-determined step (PDS) of O-O bond formation on the CoFeOOH/CeO_{2-x}N_x, which mediates O* to react with another OH⁻ to form the HOO* intermediate. This is different from the bare CoFeOOH, which undergoes the PDS of the deprotonation of *OH for the formation of *O. Owing to the influence of Ce-N bonds, the Co atoms at the CoFeOOH/CeO_{2-x}N_x interface have lower adsorption energies of *OH, *O and *OOH intermediates, giving rise to the ΔG_{PDS} value of as low as ~0.44 eV, in sharp contrast with that of amorphous CoFeOOH (~0.87 eV). The low energy barrier enlists the CoFeOOH/CeO_{2-x}N_x interface as the electroactive sites to substantially boost OER kinetics.

10. The benchmark of OER performance is RuO₂ on a glassy carbon electrode with organic binder. Yet the working electrode for CoFe/Co-O-N is a self-supported manner. This comparison was not fairly done due to the huge contact resistance for the RuO₂ case. The authors should alloy the RuO₂ with a conductive electrode (e.g. Ni foam) at high temperature to give a fair comparison. Of course, no Nafion binder should be used to block the active sites of RuO₂.

Reply: We thank the reviewer for the insightful comment and constructive suggestion, according to which we have prepared Ni foam supported RuO₂ (RuO₂/NF) by electrodepositing RuO₂ on self-supported Ni foam and then annealing at 200 °C. The OER polarization curve of RuO₂/NF has shown in **Figure 3a** and its electrochemical properties have been compared with nanoporous FeCo/CeO_{2-x}N_x, FeCo/Ce-O, FeCo, CeO_{2-x}N_x and RuO₂/GC in **Figure 3b-d**. As shown in **Figure 3a-d**, the nanoporous FeCo/CeO_{2-x}N_x exhibits superior electrocatalytic behavior compared with RuO₂/NF.

11. The impedance of Fig. 3d is confusing. How come the mass transport parts for all the critical samples are missing? This is important because the ECSA seems to be a critical impact on the OER activity (56 times higher).

Reply: We appreciate the reviewer for the constructive suggestion. Following this suggestion, we have reanalyzed the EIS spectra of nanoporous FeCo/CeO_{2-x}N_x, FeCo/Ce-O, FeCo and CeO_{2-x}N_x electrodes as well as RuO₂/NF and

RuO₂/GC based on new equivalent electrical circuit with the charge transfer resistance (R_{CT}) and the pore resistance (R_P) in parallel with the constant phase elements (CPEs), in addition to the intrinsic electrode and electrolyte resistance (R_1). According to the equivalent circuit with these general descriptors (inset of **Figure 3d**), the R_1 , R_{CT} and R_P values of all these electrodes are compared in Supplementary **Figure 16**. As shown in Supplementary **Figure 16c** for the R_P reflecting the mass transport, the nanoporous FeCo/CeO_{2-x}N_x is ~1.6 Ω , much lower than the nanoporous CeO_{2-x}N_x (~12.6 Ω).

12. It is really amazing that using a carbon rod as the HER electrode can yield such a great OER current. The reviewer would say the LSV data provided by the authors are even much better than using Pt as a HER electrode in many other works in the field. It is known that carbon rod is much more inert, and not even have a chance to compare with Pt. The geometry dimension (width, length, and height) of the self-support electrode of CoFe/Co-O-N should be provided to answer the concern regarding such the high current density.

Reply: We thank the reviewer for the comment. Following this suggestion, we have provided the geometry dimension (width, length and height) of the self-supported nanoporous FeCo/CeO_{2-x}N_x.

13. In Fig. 4, the stability tests. Please use elemental leaching in the electrolyte to support the argument for the high stability of OER at high currents. The EDS only tells you how many much the quantity of active elements remains there before leaching out. In addition, please also provide the appearance change of the working electrode before and after the 1000 hour test. Shall there be a significant color change of the electrocatalyst, provide an explanation about that.

Reply: We thank the reviewer for constructive suggestion. Following this suggestion, we have performed ICP-OES measurements of the electrolyte when performing the durability test for 100, 200, 300 and 400 hours, respectively. However, the concentrations of Fe, Co and Ce ions in the tested electrolyte are too low to be detected by ICP-OES even after the durability test for 400 hours. At the same time, we also have conducted additional characterization on the deposited metals on the counter electrode of carbon rod. There is only Fe to be detected. According to ICP-OES measurement, the specific mass of Fe electrodeposited on the carbon rod is determined to be only ~0.0116 mg cm⁻² during the OER durability test for 400 hours. If the deposited Fe derives from the dissolution of Fe of nanoporous FeCo/CeO_{2-x}N_x, the dissolution rate is 0.477 $\mu\text{g h}^{-1}$, much lower than other Fe-based electrocatalysts. This is probably due to the hierarchical nanoporous structure that ensures fast transportation of OH⁻ to keep a stable environment at active sites, lowering the

dissolution rate of metal ions. As a consequence, the nanoporous FeCo/CeO_{2-x}N_x electrode does not display evident changes in appearance and color after the durability test for 400 hours (Supplementary **Figure 17**). This accounts for the slight composition evolution of nanoporous FeCo/CeO_{2-x}N_x electrode after the durability test for 1000 hours (Supplementary **Figure 18**). Nevertheless, it still maintains initial laminate heterostructure of alternating nanoporous FeCo and CeO_{2-x}N_x lamellas with sturdy interfaces (**Figure 4c** and inset of **Figure 4b**) and stable chemical states of surface Fe, Co and Ce atoms (Supplementary **Figure 19**) due to its exceptional steady heterostructure. As a result, nanoporous FeCo/ CeO_{2-x}N_x electrode exhibits the stable current density of ~1900 mA cm⁻² even through some accidents such as power break and switch trip take place during the durability test for 1000 hours (**Figure 4b**). These results indicate the outstanding durability of nanoporous FeCo/CeO_{2-x}N_x electrode under very violent O₂ gas evolution.

Reviewer #3 (Remarks to the Author):

This work by Zeng et al. reported a laminate composite electrode composed of alternating hierarchical nanoporous bimetallic iron-cobalt alloy/cerium oxynitride (FeCo/Ce-O-N) heterolamellas as self-supported electrocatalytic material for oxygen evolution reaction. Owing to the unique architecture that offer abundant electroactive sites and facilitate electron transfer and ion transportation, the nanoporous FeCo/Ce-O-N electrode delivers ultrahigh current densities at low overpotentials and retains exceptional stability at high current densities for more than 1000 hours. This electrode is an attractive OER catalyst and holds a promise for practical application in large-scale hydrogen production via electrochemical water-splitting technologies. The manuscript is well written with clear logic. This work is novel and important to the field. I would recommend its publication in Nature Communications after the authors addressing the following points:

Reply: We appreciate the reviewer for finding interest of our work. We also appreciate the reviewer for his/her insightful comments and constructive comments. Following these comments/suggestions, we have carried out supplementary experiments on reproducibility of nanoporous FeCo/CeO_{2-x}N_x electrodes and provided additional SEM-EDS spectrum of precursor alloys. Based on these results, we have completely revised the manuscript. The detailed corrections are listed below.

(1) In Figure 1h, the authors should explain why there observe H₂O and HO in the XPS spectrum of O 1s for the as-annealed nanoporous FeCo/Ce-O-N specimen.

Reply: We appreciate the reviewer for this insightful comment. According to this comment, we have reanalyzed the XPS spectrum of O 1s for the as-annealed nanoporous FeCo/CeO_{2-x}N_x electrode. The corrected O 1s XPS spectrum has been shown in **Figure 1h**.

(2) In view that the authors mention that the current density of nanoporous FeCo/Ce-O-N can reach as high as >3900 mA cm⁻², they should extend the current density in Figure 3a.

Reply: Following this constructive suggestion, we have extended the current density to 4000 mA cm⁻² in **Figure 3a**.

(3) The authors should perform supplementary experiment to demonstrate the reproducibility of FeCo/Ce-O-N hybrid electrode.

Reply: We thank the reviewer for this constructive suggestion. Following this suggestion, we have prepared five nanoporous FeCo/CeO_{2-x}N_x electrodes by the same

alloying/dealloying and nitridation procedures, and measured their corresponding OER polarization curves in 1 KOH electrolyte. As shown in Supplementary **Figure 14**, these nanoporous FeCo/CeO_{2-x}N_x electrodes exhibit almost overlapping OER polarization curves, demonstrating the high reproducibility.

(4) The authors should provide EDS spectra of precursor alloys to demonstrate their atomic ratios of Fe, Co, Ce, Al.

Reply: Following this suggestion, we have provided EDS spectra of precursor alloys in Supplementary **Figure 1**.

(5) Some recently published related articles are suggested to be cited, for examples, Nature communications, 2022, 13, 2191; Nature communications, 2020, 11, 1664; Electrochem. Energy Rev. 2021, 4(1), 136–145; Nano-Micro Letters, 2022, 14, 120; Electrochem. Energy Rev. 2021, 4(3), 566–600; SusMat, 2021, 1(4): 460-481; Chem. Soc. Rev., 2022,51, 4583-4762; etc.

Reply: According to this suggestion, we have mentioned these papers in references for a comprehensive introduction.

REVIEWER COMMENTS

Reviewer #1 (Remarks to the Author):

Overall, I am not convinced by the reports on the origin of catalytic activity and stability.

1) High OER activity as a result of the FeOOH/CeON interface. No results that would directly confirm this claim are given in this revision. In addition, the number of active sites at the interface is small compared to the whole electrode. Consequently, it is hard to believe that those active sites alone would support the OER at high current density.

2) High OER activity is a consequence of the nonporous structure and increased conductivity. Later is doubtful as CoFeOOH is an intrinsically poor conductor, the origin of improved conductivity should be explained.

3) Authors state that CoFeOOH is stable under an oxidative OER environment. This is in contradiction with the available literature and is not in agreement with Pourbaix diagrams. If such statements are made, the origin of stability should be studied and compared with previous work that shows Fe leaching. Since Fe leaching is not observed, it should be explained how Fe is stabilized. The statement 'This is probably due to the hierarchical nanoporous structure that ensures fast transportation of OH⁻ to keep a stable environment at active sites.' is too speculative. It is hard to believe that nanoporous morphology would counter diffusion-related issues.

Reviewer #2 (Remarks to the Author):

In the revision, the reviewer would not agree that all the concerns have been correctly addressed. Thus, the reviewer would not recommend to be accepted in the present form.

The theoretical calculation (DFT) into the amorphous modes are not clear how it was done. How was the model reasonably constructed? As other reviewer suggested, is it so necessary to have DFT to support the article?

Reviewer #3 (Remarks to the Author):

I am satisfied with the revised manuscript and also the authors' reply to the reviewers' comments. I am happy to recommend its acceptance as is.

Response To Reviewers' Comments

Reviewer #1 (Remarks to the Author):

Overall, I am not convinced by the reports on the origin of catalytic activity and stability.

Reply: We appreciate the reviewer for his/her further insightful comments on catalytic activity and stability. According to the comments on the catalytic activity, we have additionally evaluated the specific activities of nanoporous electrodes and performed systematic DFT simulation to demonstrate the origin of catalytic activity of FeCoOOH/CeO_{2-x}N_x heterostructure in nanoporous FeCo/CeO_{2-x}N_x electrode. As for the stability, we have double-checked the experiments of durability test and additionally performed durability tests on nanoporous FeCo/CeO₂ and nanoporous FeCo electrodes. There indeed takes place Fe dissolution on these nanoporous electrodes under the oxidative OER environment although these dissolved Fe ions transport to the counter electrode and electrodeposit on the carbon rod during the stability test. According to the masses of Fe electrodeposition, the dissolution rates of Fe in nanoporous FeCo/CeO_{2-x}N_x, FeCo/CeO₂ and FeCo electrodes are estimated to be ~0.477 μg h⁻¹, ~0.782 μg h⁻¹, ~2.412 μg h⁻¹, respectively, reflecting the significant role of CeO_{2-x}N_x in improving the electrochemical durability. On the basis of the DFT simulation and XPS characterization, the outstanding durability of nanoporous FeCo/CeO_{2-x}N_x electrode is due to evident electron transfer from CeO_{2-x}N_x to CoFeOOH at their heterointerface, which substantially stabilizes the chemical states of Fe component to alleviate the Fe leaching that is usually caused by the formation of soluble FeO₄²⁻ under OER environment. Based on these results, we have further revised our manuscript. The detailed corrections are listed below.

(1) High OER activity as a result of the FeOOH/CeON interface. No results that would directly confirm this claim are given in this revision. In addition, the number of active sites at the interface is small compared to the whole electrode. Consequently, it is hard to believe that those active sites alone would support the OER at high current density.

Reply: We thank the reviewer for the constructive comment. Following this comment, we have additionally evaluated the specific activities of nanoporous electrodes, including nanoporous FeCo/CeO_{2-x}N_x, FeCo/Ce-O, FeCo, CeO_{2-x}N_x and RuO₂/NF, based on their electrochemical surface areas (ECSAs) calculated according to double-layer capacitance measurements. As shown in Supplementary **Figure 19**, the nanoporous FeCo/CeO_{2-x}N_x electrode has the specific activity of as high as ~558 mA cm⁻²_{ECSA} at overpotential of 360 mV. This is about one order of magnitude higher than that of nanoporous FeCo/Ce-O electrode (~0.050 mA cm⁻²_{ECSA}) due to the presence of

CoFeOOH/CeO_{2-x}N_x heterostructure as electroactive sites. This observation also demonstrates the important role of CeO_{2-x}N_x, instead of CeO₂, in improving the electrocatalytic activity of CoFeOOH via the formation of CoFeOOH/CeO_{2-x}N_x heterostructure. Because of the presence of CeO_{2-x}N_x, the specific activity of nanoporous FeCo/CeO_{2-x}N_x electrode more than 15-fold higher than that of nanoporous FeCo electrode with the electroactive CoFeOOH (~0.034 mA cm⁻²_{ECSA}) (Supplementary **Figure 19**). At the same time, we have carried out systematically DFT simulation to demonstrate the origin of electroactivity. Bader charge analysis elucidates the change in the electronic structure after the incorporation of N. As shown in Supplementary **Figure 13a,b**, the Co in CoFeOOH/FeCo/CeO_{2-x}N_x possesses a low atomic charge of +1.18|e| compared the Co in CoFeOOH/CeO₂ (+1.24|e|) and bare CoFeOOH (+1.45|e|). This will weaken the adsorption energy of *OH on CoFeOOH/FeCo/CeO_{2-x}N_x as a consequence of downshift of d-band center relative to CoFeOOH/CeO₂ and CoFeOOH (Supplementary **Figure 14**). Therefore, the CoFeOOH/CeO_{2-x}N_x mediated OER suffers from the rate-determining step (RDS) of O-O bond formation via *O intermediate reacting with another OH⁻ to form the *OOH, different from the RDSs of CoFeOOH/CeO₂ (the deprotonation of *OH for the formation of *O) and amorphous CoFeOOH (the O₂ desorption). Therein, the ΔG_{RDS} value of CoFeOOH/CeO_{2-x}N_x is as low as ~0.44 eV, in sharp contrast with those for CoFeOOH/CeO₂ (~0.61 eV) and amorphous CoFeOOH (~0.68 eV).

Besides, we also agree with reviewer that the number of active sites at the interfaces of nanoporous FeCo and CeO_{2-x}N_x lamellas is small compared with the whole electrode. According to this comment, we have double-checked SEM-EDS elemental mapping of nanoporous FeCo/CeO_{2-x}N_x electrode (**Figure 1g**). There observe some Ce atoms along with O and N to distribute in the constituent nanoporous FeCo alloy/oxide lamellas. This is probably due to the residual Ce component in the FeCo alloy lamellas caused by solidification reaction of precursor alloy. These Ce atoms remain in nanoporous FeCo alloy/oxide lamellas during the chemical dealloying and thermal nitridation, enabling the further formation of more CoFeOOH/CeO_{2-x}N_x heterointerfaces in the OER (Supplementary **Figure 5**), in addition to the ones located between nanoporous FeCo alloy/oxide and CeO_{2-x}N_x lamellas. Therefore, it is reasonable to attribute the outstanding electrocatalytic behaviors of nanoporous FeCo/CeO_{2-x}N_x electrode to the unique CoFeOOH/CeO_{2-x}N_x heterostructure interface.

(2) High OER activity is a consequence of the nonporous structure and increased conductivity. Later is doubtful as CoFeOOH is an intrinsically poor conductor, the origin of improved conductivity should be explained.

Reply: We appreciate the reviewer for the insightful comment. According to this

comment, we have explained the origin of improved conductivity in the revised manuscript. As demonstrated by Raman spectrum (**Figure 2b**), the weak intensity of the characteristic Raman band at $\sim 590\text{ cm}^{-1}$ indicates that there only form ultrathin CoFeOOH layer on the surface of nanoporous FeCo alloy/oxide ligaments after initial OER test. Although the CoFeOOH is intrinsically of poor conductivity, the R_1 value (corresponding to intrinsic resistance of both electrolyte and electrode) of nanoporous FeCo/CeO_{2-x}N_x electrode is as low as $\sim 4.5\ \Omega$ (Supplementary **Figure 20c**). This is probably due to the unique architecture of ultrathin CoFeOOH layer in-situ forming on the surface of interconnective conductive FeCo alloy skeleton, which is conducive to electron transportation during the OER processes.

(3) Authors state that CoFeOOH is stable under an oxidative OER environment. This is in contradiction with the available literature and is not in agreement with Pourbaix diagrams. If such statements are made, the origin of stability should be studied and compared with previous work that shows Fe leaching. Since Fe leaching is not observed, it should be explained how Fe is stabilized. The statement 'This is probably due to the hierarchical nanoporous structure that ensures fast transportation of OH⁻ to keep a stable environment at active sites.' is too speculative. It is hard to believe that nanoporous morphology would counter diffusion-related issues.

Reply: We appreciate the reviewer for this insightful comment. Owing to the fast consumption rate of OH⁻ and the localized pH change, there unavoidably takes place Fe leaching, as reported previously. In our durability measurements of nanoporous FeCo/CeO_{2-x}N_x, as well as the additional nanoporous FeCo/CeO₂ and nanoporous FeCo, there indeed occurs Fe dissolution via the formation of soluble FeO₄²⁻ under OER environment. These dissolved Fe ions transport to counter electrode and electrodeposit on carbon rod. Therefore, there do not detect any Fe, Co and Ce ions in the electrolyte at the different test time by inductively coupled plasma optical emission spectroscopy (ICP-OES) but the trace amount of Fe on carbon rod (Supplementary **Table 2**). According to the masses of Fe electrodeposition, the dissolution rate of Fe in nanoporous FeCo/CeO_{2-x}N_x, FeCo/CeO₂ and FeCo electrodes are estimated to be $\sim 0.477\ \mu\text{g h}^{-1}$, $\sim 0.782\ \mu\text{g h}^{-1}$ and $\sim 2.412\ \mu\text{g h}^{-1}$, respectively. These observations reflect the significant role of CoFeOOH/CeO_{2-x}N_x heterostructure in improving the electrochemical stability of nanoporous FeCo/CeO_{2-x}N_x electrode probably due to evident electron transfer from CeO_{2-x}N_x to CoFeOOH at their heterointerface (Supplementary **Figure 13a**), which substantially stabilizes the chemical states of Fe component to alleviate the formation of soluble FeO₄²⁻ for Fe leaching under OER environment. Specifically, the surface Fe component has the Fe³⁺/Fe²⁺ ratio of 70.7:29.3 after the durability test for 1000 hours (Supplementary **Figure 23**), the almost

same as the initial value of 70.3: 29.7 (Supplementary **Figure 10b**). As a consequence, nanoporous FeCo/CeO_{2-x}N_x electrode exhibits the stable current density of ~1900 mA cm⁻² even through some accidents such as power break and switch trip take place during the durability test for 1000 hours, in sharp contrast with nanoporous FeCo/CeO₂ and nanoporous FeCo ones, which encounter remarkable reduction in current densities in 100 hours (**Figure 4b**).

Reviewer #2 (Remarks to the Author):

In the revision, the reviewer would not agree that all the concerns have been correctly addressed. Thus, the reviewer would not recommend to be accepted in the present form.

Reply: We appreciate the reviewer for his/her further comment and suggestion, according to which we have supplemented the detailed method to construct amorphous atomic model and additionally performed DFT simulation on electronic properties. The detailed correction is listed below.

The theoretical calculation (DFT) into the amorphous modes are not clear how it was done. How was the model reasonably constructed? As other reviewer suggested, is it so necessary to have DFT to support the article?

Reply: We thank the reviewer for this insightful comment. Following this comment, we have supplemented the detailed method to construct amorphous model of CoFeOOH/CeO_{2-x}N_x in Method section. We first established 2 × 4 × 1 supercell of CoFeOOH(001) and 3 × 2 × 2 one of CeO₂(111) after their ($\sqrt{3}\times 1$)R30° reconstructions with the lattice angles adjusting to 90°. Therein, the dimensions of CoFeOOH(001) and CeO₂(111) supercells are 12.97 Å × 11.24 Å and 13.25 Å × 11.48 Å, respectively. To construct the amorphous structure, ab initio molecule dynamics (AIMD) calculations were performed on CoFeOOH(001) slab under the NVT condition at 300 K, based on experimental condition. The total simulation time was 5 ps with a time step of 1 fs (5000 simulation steps). After the formation of amorphous CoFeOOH, we further constructed atomic models of CoFeOOH/CeO₂ heterostructures with or without N doping by combining the amorphous CoFeOOH layer and the N-undoped or doped Ce terminal CeO₂(111) slab. The vacuum region of 15 Å was set to avoid the interaction between periodic images. Notably, each step in these model constructions was accompanied with a geometry optimization calculation, wherein the convergence criterions of the energy and force were set to 1×10⁻⁵ eV and 0.02 eV Å⁻¹, respectively. Based on these atomic models, we have additionally provided charge density difference diagram and partial density of state to demonstrate the influence of CeO_{2-x}N_x on the electroactive activity and stability of CoFeOOH via electron transfer.

Reviewer #3 (Remarks to the Author):

I am satisfied with the revised manuscript and also the authors' reply to the reviewers' comments. I am happy to recommend its acceptance as is.

Reply: We appreciate the reviewer for recommend our manuscript for publication in *Nature Communications*.

Response To Reviewers' Comments

Reviewer #1 (Remarks to the Author):

Reply: We appreciate the reviewer for his/her recommending our manuscript for publication in *Nature Communications*.

Reviewer #2 (Remarks to the Author):

We appreciate the reviewer for his/her recommending our manuscript for publication in *Nature Communications*.